

# The Hestia Fossil Fuel $CO_2$ Emissions Data Product for the Los
# Angeles Megacity (Hestia-LA)
Kevin R. Gurney[1], Risa Patarasuk[4], Jianming Liang[2,3], Yang Song[2], Darragh O'Keeffe[5], Preeti
Rao[6], James R. Whetstone[7], Riley M. Duren[8], Annmarie Eldering[8], Charles Miller[8]
[1]School of Informatics, Computing, and Cyber Systems, Northern Arizona University, Flagstaff, AZ, USA
[2]School of Life Sciences, Arizona State University, Tempe AZ USA
[3]Now at ESRI, Redlands, CA USA
[4]Citrus County, Dept. of Systems Management, Lecanto, FL, USA
[5]Contra Costa County, Department of Information Technology, Martinez, CA, USA
[6]School for Environment and Sustainability, University of Michigan, Ann Arbor, MI, USA
[7]National Institute for Standards and Technology, Gaithersburg, MD, USA
[8]NASA Jet Propulsion Laboratory, California Institute of Technology, Pasadena, CA, USA
*Correspondence to*: Kevin R. Gurney (kevin.gurney@nau.edu)
**Abstract**. As a critical constraint to atmospheric $CO_2$ inversion studies, bottom-up spatiotemporally-explicit
emissions data products are necessary to construct comprehensive $CO_2$ emission information systems useful for
trend detection and emissions verification. High-resolution bottom-up estimation is also useful as a guide to
mitigation options, offering details that can increase mitigation efficiency and synergize with other policy goals at
the national to sub-urban spatial scale. The 'Hestia Project' is an effort to provide bottom-up fossil fuel ($FFCO_2$)
emissions at the urban scale with building/street and hourly space-time resolution. Here, we report on the latest
urban area for which a Hestia estimate has been completed – the Los Angeles Megacity, encompassing five
counties: Los Angeles County, Orange County, Riverside County, San Bernardino County and Ventura County. We
provide a complete description of the methods used to build the Hestia $FFCO_2$ emissions data product which is
presented on a 1 km x 1 km grid for the years 2010-2015. We find that the LA Basin emits 48.06 ($\pm$ 5.3) MtC/yr,
dominated by the onroad sector. Because of the uneven spatial distribution of emissions, 10% of the largest emitting
gridcells account for 93.6%, 73.4%, 66.2%, and 45.3% of the industrial, commercial, onroad, and residential sector
emissions, respectively. Hestia $FFCO_2$ emissions are 10.7% larger than the inventory estimate generated by the local
metropolitan planning agency, a difference that is driven by the industrial and electricity production sectors. The
Hestia-LA v2.5 emissions data product can be downloaded from the data repository at the National Institute of
Standards and Technology (https://doi.org/10.18434/T4/1502503).
## 1 Introduction
Driven by the growth of fossil fuel combustion, the amount of carbon dioxide ($CO_2$), the most important
anthropogenic greenhouse gas (GHG) in the Earth's atmosphere, recently reached an annual average global mean
concentration of $402.8 \pm 0.1$ parts per million (ppm) on its way to doubling pre-industrial levels (IPCC, 2013;
LeQuere et al., 2018). We have also witnessed the first time that the majority of world's inhabitants reside in urban
areas. This trend, like atmospheric $CO_2$ levels, is intensifying. Projections show cities worldwide could add 2 to 3
billion people this century and are projected to triple in area by 2030 (UN DESA 1015; Seto et al., 2012).

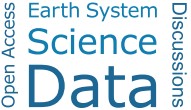

These two thresholds are linked—almost three-quarters of energy-related, atmospheric $CO_2$ emissions are driven by
urban activity (Seto et al., 2014). If the world's top 50 emitting cities were counted as one country, that nation would
rank third in emissions behind China and the United States (World Bank 2010). Indeed, urbanization is a factor
shaping national contributions to internationally agreed emission reductions, as subnational governments are playing
an increasing role in climate mitigation and adaptation policy implementation (Bulkeley 2010; Hsu et al., 2017).
Furthermore, the pace of urbanization continues to increase and opportunities to avoid carbon "lock-in" - where
relationships between technology, infrastructure, and urban form dictate decades of high-$CO_2$ development - are
diminishing (Ürge-Vorsatz et al., 2018; Seto et al., 2016; Erickson et al., 2015).
Motivated by these numerical realities and the recognition that low-emission development is consistent with a
variety of other co-benefits (e.g. air quality improvement), cities are taking steps to mitigate their $CO_2$ emissions
(Rosenzweig et al., 2010; Hsu et al., 2015; Watts 2017). For example, 9120 cities representing over 770 million
people (10.5% of global population) have committed to the Global Covenant of Mayors (GCoM) to promote and
support action to combat climate change (GCoM, 2018). Over 90 large cities, as part of the C40 network, have
similarly committed to mitigation actions with demonstrable progress. However, the scale of actual reductions
remains modest, despite the many pledges and initial progress. For example, a recent study reviewed 228 cities
pledged to reduce 454 megatons of $CO_2$ per year by 2020 (Erickson and Lazarus, 2012). Were they to meet these
commitments, the reduction would account for about 3% of current global urban emissions and less than 1% of total
global emissions projected for 2020. More important, there is a need for timely information to manage and assess
the performance of implemented mitigation efforts and policies (Bellassen et al., 2015).
One of the barriers to targeting a deeper list of emission reduction activities is the limited amount of actionable
emissions information at scales where human activity occurs: individual buildings, vehicles, parks, factories and
power plants (Gurney et al., 2015). These are the scales at which interventions in $CO_2$-emitting activity must occur.
Hence, the emissions magnitude and driving forces of those emissions must be understood and quantified at the
"human" scale to make efficient (i.e. prioritizing the largest available emitting activities/locales) mitigation choices
and to capture the urban co-benefits that also occur at this scale (e.g. improve traffic congestion, walkability, green
space). Similarly, a key obstacle to assessing progress is a lack of independent atmospheric evaluation (ideally
consistent in space and time with the human-scale emissions mapping) (Duren and Miller 2011).
Existing methods and tools to account for urban emissions have been developed primarily in the non-profit
community (WRI/WBCSD, 2004; Fong et al., 2014). In spite of these important efforts, most cities lack
independent, comprehensive and comparable sources of data and information to drive and/or adjust these
frameworks. Furthermore, the existing tools and methods are designed at an aggregate level (i.e. whole city, whole
province), missing the most important scale—sub-city—and hence provide limited actionable information.
The scientific community has begun to build information systems aimed at providing independent assessment of
urban $CO_2$ emissions. Through a combination of atmospheric measurements, atmospheric transport modeling and
data-driven "bottom-up" estimation, the scientific community is exploring different methodologies, applications,
and uncertainty estimation of these approaches (Hutyra et al., 2014). Atmospheric monitoring includes ground-based
$CO_2$ concentration measurements (McKain et al., 2012; Djuricin et al., 2010; Miles et al., 2017; Turnbull et al.,



2015, Verhulst et al., 2017), ground-based eddy flux measurements (Christen 2014; Crawford and Christen 2014;
Grimmond et al., 2002; Menzer et al., 2015; Velasco and Roth 2010; Velasco et al., 2005), aircraft-based flux
measurements (Mays et al., 2009; Cambaliza et al., 2014; 2015) and whole-column abundances from both ground,
and space-based, remote sensing platforms (Wunch et al., 2009; Kort et al., 2012; Wong et al., 2015; Schwandner et
al., 2018).
"Bottom-up" approaches, by contrast, include a mixture of direct flux measurement, indirect measurement and
modeling. Common among the bottom-up approaches are those that include flux estimation based on a combination
of activity data (population, number of vehicles, building floor area) and emission factors (amount of $CO_2$ emitted
per activity), socioeconomic regression modeling, or scaling from aggregate fuel consumption (VandeWeghe and
Kennedy, 2007; Shu and Lam, 2011; Zhou and Gurney, 2011; Gurney et al., 2012; Jones and Kammen, 2014;
Ramaswami and Chavez, 2013; Patarasuk et al., 2016; Porse et al., 2016). Direct end-of-pipe flux monitoring is
often used for large point sources such as power plants (Gurney et al., 2016). Indirect fluxes (those occurring outside
of the domain of interest but driven by activity within) can be estimated through either direct atmospheric
measurement (and apportioned to the domain of interest) or can be modeled through process-based (Clark and
Chester 2017) or economic input-output models (Ramaswami et al., 2008).
Integration of bottom-up urban flux estimation with atmospheric monitoring has been achieved with atmospheric
inverse modeling, an approach whereby surface fluxes are estimated from a best fit between bottom-up estimation
and fluxes inferred, via atmospheric transport modeling, from atmospheric concentrations (Lauvaux et al., 2013;
Lauvuax et al., 2016; Breon et al., 2015; Davis et al., 2017). Though the various measurement and modeling
components continue to be tested, integration offers an urban anthropogenic $CO_2$ information system which can
provide accuracy, emissions process information, and spatiotemporal detail. This combination of attributes satisfies
a number of urgent requirements. For example, it can offer the means to evaluate urban emissions mitigation efforts
by assessing urban trends. Space, time, and process detail of emitting activity can guide mitigation efforts,
illuminating where efficient opportunities exist to maximize reductions or focus new efforts. Finally, emissions
quantification is also seen as a potentially powerful metric with which to better understand the urbanization process
itself, given the importance of energy consumption to the evolution of cities.
The Hestia Project was begun to estimate bottom-up urban fossil fuel $CO_2$ (FFCO$_2$) fluxes for use within integrated
flux information systems. Begun in the city of Indianapolis, the Hestia effort is now part of a larger experiment that
includes many of the modeling and measurement aspects described above. Referred to as the Indianapolis Flux
Experiment (INFLUX), this integrated effort has emerged to test and explore quantification and uncertainties of the
urban $CO_2$ and $CH_4$ measurement and modeling approaches using Indianapolis as the testbed experimental
environment (Whetstone et al., 2018; Davis et al., 2017).
Because urban areas differ in key attributes such as size, geography, and emission sector composition, multiple cities
are now being used to test aspects of anthropogenic $CO_2$ monitoring and modeling. The Hestia approach has been
used in a number of these urban domains. Here, we provide the methods and results from one of those urban domains,
the Los Angeles Basin Megacity. The Hestia-LA effort was developed under the Megacities Carbon framework
(https://megacities.jpl.nasa.gov/portal/). It was designed to serve the Megacities Carbon Project in a similar capacity



to its role in INFLUX. The Hestia-LA result is unique in that it is the first high-resolution spatiotemporally-explicit inventory of $FFCO_2$ emissions centered over a megacity. A preliminary version of Hestia-LA containing only the transportation sector emissions was reported by Rao et al. (2017). While emphasis thus far has been focused on atmospheric $CH_4$ monitoring analyses in the LA megacity (Carranza et al., 2017; Wong et al., 2016; Verhulst et al., 2017; Hopkins et al., 2016), work is ongoing to use the extensive atmospheric $CO_2$ observing capacity in the Los Angeles domain (e.g. Newman et al., 2016; Feng et al., 2016; Wong et al., 2015; Wunch et al., 2009) within an atmospheric $CO_2$ inversion.

In this paper, we describe the study domain, the input data, uncertainty, and the methods used to generate the Hestia-LA (v2.5) data product and provide descriptive statistics at various scales of aggregation. We compare the Hestia results to the metro region planning authority estimate and place the results in the context of urban greenhouse gas mitigation. We discuss known gaps and weaknesses in the approach and goals for future work.

## 2 Methods

### 2.1 Study Domain

The Los Angeles metropolitan area is the second-largest metropolitan area in the United States and one of the largest metropolitan areas in the world. Under the definition of the Metropolitan Statistical Area (MSA) by the U.S. Office of Management and Budget, Metropolitan Los Angeles consists of Los Angeles and Orange counties with a land area of 12,562 $km^2$ and a population of 9,819,000. The Greater Los Angeles Area, as a Combined Statistical Area (CSA) defined by the U.S. Census Bureau, encompasses the three additional counties of Ventura, Riverside, and San Bernardino with a total land area of 87,945 $km^2$ and an estimated population of 18,550,288 in 2014. The Hestia-LA $FFCO_2$ emissions data product covers the complete geographic extent of these five counties including the Eastern, relatively non-urbanized portions of San Bernardino and Riverside counties. Airport emissions associated with aircraft up to 3000 feet are included as are marine shipping emissions out to 12 nautical miles from the coastal boundary.

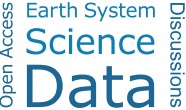

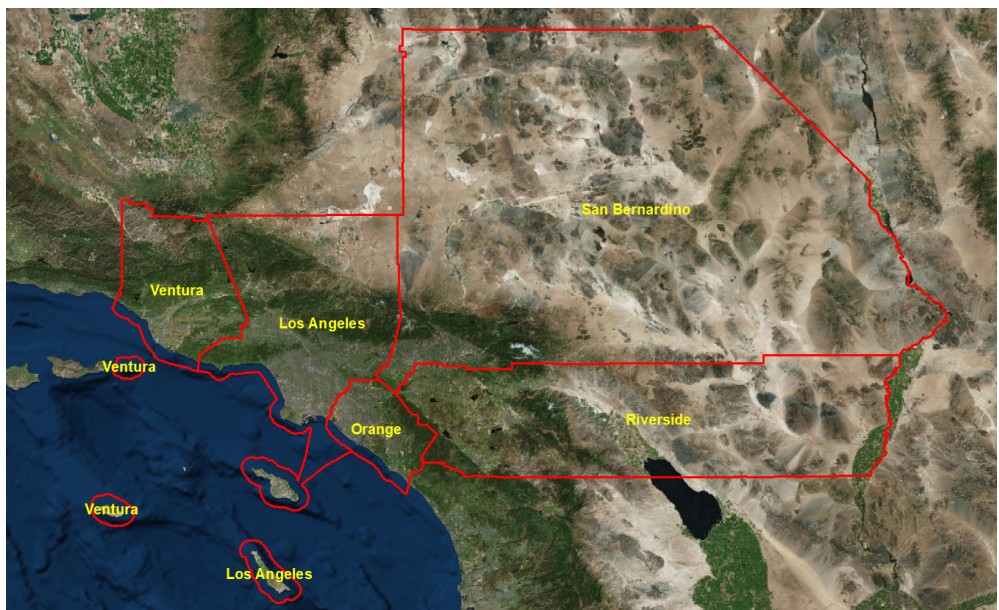

134

**Figure 1: The Hestia-LA urban domain**

## 2.2 Input data

Input data to the Hestia-LA data product are supplied by output of the Vulcan Project (Figure 2), a quantification of
$FFCO_2$ emissions at fine space and time scales for the entire US landscape (Gurney et al., 2009) The Hestia-LA
process extracts these results for the five counties within the Hestia-LA domain and adjusts these estimates where
superior local data are available and further downscales/distributes the Vulcan v3.0 results to buildings and street
segments. Details of the Vulcan v3.0 methodology is provided elsewhere (Gurney et al., 2018). Here, we summarize
the Vulcan v3.0 methods and then provide greater detail regarding the Hestia-LA processing of that data to high-
resolution space/time scales.

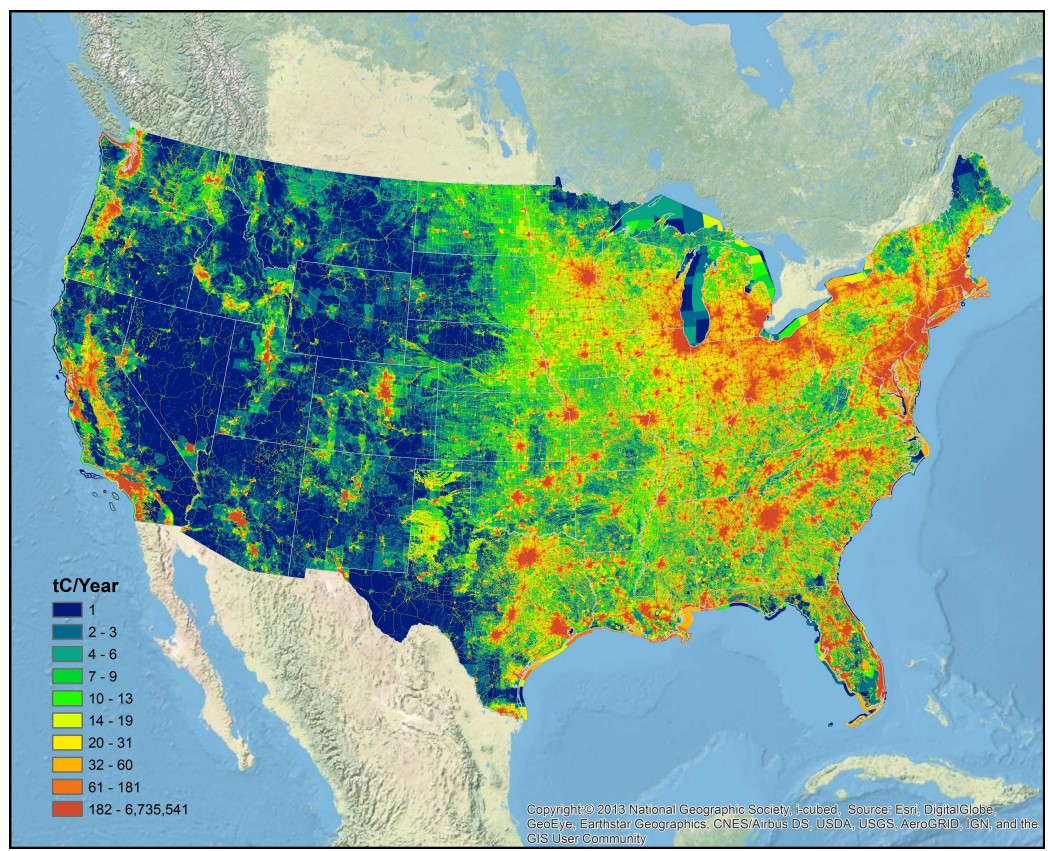


**Figure 2: Total annual FFCO$_2$ emissions for the year 2011 from the Vulcan v3.0 output.**
The Vulcan v3.0 input data (the output of which is the input for the Hestia-LA) are organized following nine
economic sector divisions (see Table 1) - residential, commercial, industrial, electricity production, onroad, nonroad,
railroad, commercial marine vessel, and airport. Also included are emissions associated with the calcining process in
the production of cement. The data sources within each sector are either acquired as FFCO$_2$ emissions (the onroad
sector and most of the nonroad and electricity production sectors) or as carbon monoxide (CO) emissions (all other
sectors) and transformed to FFCO$_2$ emissions via emission factors. Furthermore, the data sources are represented
geographically as either geocoded emitting locations ("point") or as spatial aggregates ("nonpoint" or area-based
emissions). Point sources are stationary emitting entities identified to a geocoded location such as industrial facilities
in which emissions exit through a stack or identifiable exhaust feature (USEPA, 2015a). Area or nonpoint source
emissions are not inventoried at the facility-level but represent diffuse emissions within an individual U.S. county.
Because the focus of the current study is CO$_2$ emissions resulting from the combustion of a fossil fuels, fugitive or
evaporative emissions are not included nor are "process" emissions, for example, associated with high-temperature
metallurgical processes.
Much of the input data for Vulcan v3.0 are acquired from the Environmental Protection Agency's (EPA) National
Emission Inventory (NEI) for the year 2011 (referred to hereafter as the "2011 NEI") which is a comprehensive





inventory of all criteria air pollutants (CAPs) and hazardous air pollutants (HAPs) across the United States (USEPA,
2015b). All of the individual record-level reporting in the 2011 NEI comes with a source classification code (SCC)
which codifies the general emission technology, fuel type used, and sector (*USEPA* 1995).
$FFCO_2$ emissions from the electricity production sector are primarily retrieved from two sources other than the 2011
NEI. The first is the EPA's Clean Air Markets Division (CAMD) data (USEPA, 2015c) which reports $FFCO_2$
emissions at geocoded electricity production facility locations. The second is the Department of Energy's Energy
Information Administration (DOE EIA) reporting data (DOE/EIA, 2003) which reports fuel consumption at
geocoded electricity production facility locations. Some electricity production emissions are retrieved from the 2011
NEI (as CO emissions). Overlap between these three data sources is eliminated via preference in the order listed
above. A detailed comparison made between the CAMD and EIA $FFCO_2$ emissions along with greater detail
regarding data sources, data processing and procedures can be found in Quick et al., (2014) and Gurney et al. (2014;

172 2016; 2018).

The 2011 onroad $FFCO_2$ emissions are retrieved from the EMissions FACtors 2014 model (EMFAC2014), produced
by the California Air Resources Board (CARB 2014). Onroad transportation represents all mobile transport using
paved roadways and include both private and commercial vehicles of many individual classes (e.g., passenger
vehicles, buses, light duty trucks, etc). The nonroad sector, by contrast, includes all surface mobile vehicles that do
not travel on designated paved roads surface and include a large class of vehicles such as construction equipment
(e.g., bulldozers, backhoes, etc.), ATVs, snowmobiles, and airport fueling vehicles. The nonroad emissions are
derived from the 2011 NEI reporting of nonroad CO emissions. Airport emissions include all the emissions
emanating from aircraft during their taxi, takeoff, landing cycles up to 3000 feet and are derived from the 2011 NEI
point reporting. Other activities occurring at airports resulting in $FFCO_2$ emissions are captured in the commercial
building sector (building heating) or the nonroad sector (baggage vehicles), sourced to the 2011 NEI nonpoint, 2011
NEI point and 2011 NEI nonroad reporting. Railroad emissions include passenger and freight rail travel and are
sourced to the 2011 NEI nonpoint and point reporting. Commercial marine vessels (CMV) include all commercial-
based aquatic vessels on either ocean or freshwater sourced to the 2011 NEI nonpoint reporting. Personal aquatic
vehicles such as pleasure craft and sailboats are included in the nonroad sector. Emissions associated with cement
calcining are included given its potential size and the tradition of including it with $CO_2$ inventories and use
information from multiple sources (PCA, 2006; USGS, 2003; IPCC, 2006).
The $FFCO_2$ emissions input to the Hestia system from the Vulcan v3.0 output is associated with spatial elements
represented by points, lines and polygons, depending upon the data source, the sector and the available spatial proxy
data (Table 1). Further spatialization and temporalization occurs in the Hestia system.
**Table 1. Data sources used in the spatiotemporal distribution of $FFCO_2$ emissions (text provides acronym**
**explanations and sources).**

| Sector/type | Emissions Data Source | Original spatial resolution/information | Spatial distribution | Temporal distribution |
|---|---|---|---|---|
| Onroad | EMFAC[a], EPA NEI[b] onroad | County, road class, vehicle class | SCAG AADT[c] | PeMS[d], CCS[e] |
| Electricity production | CAMD[f] CO2, EIA[g] fuel, EPA NEI point CO | Lat/lon, fuel type, technology | EPA NEI Lat/Lon, Google Earth | CAMD, EIA and EPA |



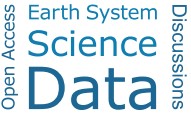

| Residential nonpoint buildings | EPA NEI nonpoint CO | County, fuel type | SCAG Parcel, floor area, DOE RECS NE-EUI[h], LA County building footprint | eQUEST[i] |
|---|---|---|---|---|
| Nonroad | NEI nonpoint CO | County, vehicle class | EPA spatial surrogates (vehicle class specific) | EPA temporal surrogates (by SCC[j]) |
| Airport | EPA NEI point CO | Lat/lon, aircraft class | Lat/Lon | LAWA[k] |
| Commercial nonpoint buildings | EPA NEI nonpoint CO | County, fuel | SCAG Parcel, floor area, DOE CBECS NE-EUI[l] | eQUEST |
| Commercial point sources | EPA NEI point CO | Lat/lon, fuel type, combustion technology | EPA NEI Lat/Lon, Google Earth | eQUEST |
| Industrial point sources | EPA NEI point CO | Lat/lon, fuel type, combustion technology | EPA NEI Lat/Lon, Google Earth | EPA temporal surrogates (by SCC) |
| Industrial nonpoint buildings | EPA NEI nonpoint CO | County, fuel type | SCAG-Parcel, floor area, DOE MECS NE-EUI[m] | eQUEST |
| Commercial Marine Vessels | EPA NEI nonpoint CO | County, fuel type, port/underway | MEM[n] | MEM |
| Railroad | EPA NEI nonpoint CO, EPA NEI point CO | County, fuel type, segment | EPA NEI rail shapefile and density distribution | EPA temporal surrogates (by SCC) |

a.   Emissions Factors Model
b.   Environmental Protection Agency, National Emissions Inventory
c.   Southern California Association of Governments, Annual Average Daily Traffic
d.   Performance Measurement System
e.   Continuous Count Stations
f.   Clean Air Markets Division
g.   Energy Information Administration
h.   Department of Energy Residential Energy Consumption Survey, non-electric energy use intensity
i.   Quick Energy Simulation Tool
j.   Source Classification Code
k.   Los Angeles World Airport
l.   Department of Energy Commercial Energy Consumption Survey, non-electric energy use intensity
m.   Department of Energy Manufacturing Energy Consumption Survey, non-electric energy use intensity
n.   Marine Emissions Model
To estimate FFCO$_2$ emissions as a multiyear time series from 2010 to 2015, the results for the year 2011 were scaled
using sector/state/fuel consumption data (thermal units) from the DOE EIA (DOE/EIA, 2018). The electricity
production sector was an exception to this approach where year-specific data was available in the CAMD and EIA
data sources. Ratios were constructed relative to the year 2011 in all SEDS sector designations for each US state.
The ratio values are applied to the annual totals in each of the sector/fuel categories specific to the state FIPS code.
**2.3   Space/time processing**
**2.3.1 Residential, commercial, industrial nonpoint buildings**
The general approach to spatializing the residential, commercial and industrial nonpoint FFCO$_2$ emissions is to
allocate the county-scale, fuel-specific annual sector totals to individual buildings (or parcels) using data on building
type, building age, total floor area, energy use intensity, and location.
A portion of the Hestia-LA building information were provided by the Southern California Association of
Governments (SCAG) (SCAG, 2012) and included building type, age, floor area, and location. The spatial
resolution of this information was at the land parcel scale (larger than the building footprint). Building footprint data
was available in the county of Los Angeles only which offered additional building floor area information needed to
correct some floor area values in the SCAG parcel data (LAC, 2016). For example, a large number of commercial
parcels with zero floor area were found in the Riverside County data which were visually inspected in Google Earth
to contain qualifying buildings. These floor area values were corrected through the combination of the Census
block-group General Building Stock (GBS) database from the Federal Emergency Management Agency (FEMA)





(FEMA, 2017) and the National Land Cover Database 2011 (NLCD) which classifies the US land surface in 30m
pixels (Homer et al., 2015).
Building energy use intensity was derived from data gathered by the DOE EIA and the California Energy
Commission (CEC). The DOE EIA Commercial Buildings Energy Consumption Survey (CBECS), Manufacturing
Energy Consumption Survey (MECS), and Residential Energy Consumption Survey (RECS) represent regional
surveys of building energy consumption categorized by building type, fuel type, and age cohort (RECS, 2013;
CBECS, 2016; MECS, 2010). Data for the Pacific West Census Division was used and in the case of the commercial
sector, was appended by the CECs Commercial End-Use Survey (CEUS) data (CEC, 2006).
In the residential sector the non-electric energy use intensity (NE-EUI) was calculated from the reported energy
consumed and total floor area sampled specific to five building types (Table 2) in the 2009 RECS survey. This was
additionally categorized by fuel type (natural gas and fuel oil) and two age cohorts (pre-1980, post-1979).
**Table 2. Residential NE-EUI survey values by building type from the Residential Energy Consumption**
**Survey (RECS)**

| RECS building type | Pre-1980 NG NE-EUI (kbtu/ft²) | Post-1979 NG NE-EUI (kbtu/ft²) | Pre-1980 Fuel oil NE-EUI (kbtu/ft²) | Post-1979 Fuel oil NE-EUI (kbtu/ft²) |
|---|---|---|---|---|
| Mobile home | 52.56 | 22.90 | NA* | NA |
| Single-family detached house | 24.53 | 18.00 | 18.87 | 7.23 |
| Single-family attached house | 42.56 | 32.38 | NA | NA |
| Apartment building with 2-4 units | 27.84 | 42.27 | NA | NA |
| Apartment building with 5 or more units | 17.21 | 30.85 | NA | NA |

* "NA" – not applicable. This indicates that there was no fuel consumption of this type evident from the survey data.
In the commercial sector, the NE-EUI was similarly calculated from the 2012 CBECS energy consumption
microdata and total floor area sampled specific to twenty building types, two fuel types (natural gas and fuel oil) and
two age cohorts (pre-1980 and post-1979). However, the sampling for the two age cohorts was insufficient to
generate estimates and the age distinction was eliminated. Furthermore, where the sample sizes remained small, NE-
EUI data from the CEUS was used in place of CBECS estimates (7 of 20 building types qualified). As the CEUS
follows a building typology different from CBECS, a crosswalk of building types between the two datasets was
necessary (Table 3).
**Table 3. Building type crosswalk and NE-EUI values for commercial buildings derived from the CBECS and**
**CUES databases**

| CBECS building class | CUES building class | NG NE-EUI (kbtu/ft²) | Fuel oil NE-EUI (kbtu/ft²) |
|---|---|---|---|
| Vacant | Miscellaneous | 9.3 | 2.5 |
| Office | All Offices | 17.9* | 1.67 |
| Laboratory | Miscellaneous | 174.7 | 0.93 |
| Nonrefrigerated warehouse | Unrefrigerated Warehouse | 3.1* | 1.03 |
| Food sales | Food Store | 27.6* | 2.5 |
| Public order and safety | Miscellaneous | 58.2 | 2.09 |
| Outpatient health care | Health | 29.1 | 3.05 |
| Refrigerated warehouse | Refrigerated Warehouse | 5.6* | 2.5 |
| Religious worship | Miscellaneous | 35.7 | 0.00 |
| Public assembly | Miscellaneous | 26.5 | 0.23 |
| Education | College, School | 25.1* | 1.7 |
| Food service | Restaurant | 210* | 100.5 |
| Inpatient health care | Health | 113.9 | 2.6 |





| Nursing | Health | 67.4 | 1.2 |
|---|---|---|---|
| Lodging | Lodging | 42.4* | 1.4 |
| Strip shopping mall | Retail | 62.7 | 2.5 |
| Enclosed mall | Retail | 4.8 | 0.02 |
| Retail other than mall | Retail | 13.6 | 16.7 |
| Service | Miscellaneous | 34.2 | 0.45 |
| Other | Miscellaneous | 18.5 | 5.3 |

* NE-EUI uses the CUES NE-EUI value due to sampling limitations in the CBECS data.
Unlike the commercial and residential survey data, the 2010 MECS survey data does not quantify energy
consumption for individually sampled buildings but rather reports the sum of the sampled buildings within each
census region categorized by manufacturing sector. The resulting NE-EUI values are shown in in Table 4. Like the
commercial data, there was inadequate sampling to justify two age cohorts.
**Table 4. Industrial NE-EUI survey values from the DOE EIA MECS database**

| MECS Class | NG NE-EUI (kbtu/ft$^2$) | Fuel oil NE-EUI (kbtu/ft$^2$) |
|---|---|---|
| Food | 519.3 | 30.5 |
| Beverage and Tobacco Products | 162.4 | 8.5 |
| Textile Mills | 144.9 | 9.3 |
| Textile Product Mills | 63.4 | 0 |
| Apparel | 35.1 | 0 |
| Leather and Allied Products | 66.7 | 0 |
| Wood Products | 76.6 | 49.5 |
| Paper | 672.8 | 69.1 |
| Printing and Related Support | 96.6 | 0 |
| Petroleum and Coal Products | 9766.0 | 436.2 |
| Chemicals | 2126.3 | 17.9 |
| Plastics and Rubber Products | 124.7 | 2.4 |
| Nonmetallic Mineral Products | 556.0 | 48.9 |
| Primary Metals | 895.0 | 16.7 |
| Fabricated Metal Products | 124.2 | 2.3 |
| Machinery | 78.6 | 3.3 |
| Computer and Electronic Products | 80.0 | 0 |
| Electrical Equip., Appliances, and Components | 133.3 | 3.7 |
| Transportation Equipment | 100.6 | 4.0 |
| Furniture and Related Products | 28.6 | 0 |
| Miscellaneous | 44.7 | 2.8 |

The NE-EUI values derived from the CBECS/RECS/MECS and CEUS survey data reflect the total building fuel
consumption for a specific fuel in a census region divided by the total floor area of all buildings in that census region
consuming that fuel. This generates a mean building NE-EUI value. Actual buildings will vary around that mean
value for a variety of reasons including different occupancy schedules, different energy efficiencies (in the envelope
or heating/cooling system), different microclimate, and other physical/behavioral characteristics. Furthermore, the
NE-EUI value applied in this way will not capture the reality that some buildings do not use fossil fuel (electricity-
only buildings) or that some buildings use one fossil fuel only versus another or use a mix of fuels in a proportion
different from the county total. Hence, each building will be allocated a mix of fossil fuel consumption identical to
the county total.





**2.3.1.1 Spatial distribution**
The county-scale commercial, residential and industrial nonpoint FFCO$_2$ emissions are allocated to each land parcel
in proportion to the product of the NE-EUI and the total floor area,
$$EC(b)_s^f = NE\_EUI_s^f \, FA(b) \tag{1}$$
where the energy consumed, $EC$, in each building, $b$, is the product of the NE-EUI value, $NE\_EUI$, and the floor
area, $FA$, for each fuel, $f$, and each building in sector, $s$. The total energy consumed, $TEC$, within the county for a
sector, $s$, is the sum of all the EC values across the $N$ buildings in the sector,
$$TEC_s^f = \sum_{b=1}^{N} EC(b)_s^f \tag{2}$$
To convert this to FFCO$_2$ emissions, we first calculate the fraction of the total energy consumption associated with
each building,
$$F(b)_s^f = \frac{EC(b)_s^f}{TEC_s^f} \tag{3}$$
where, $F$ is the fraction of $TEC$ consumed in building, $b$, of sector $s$. This is then used to distribute the county total
FFCO$_2$ emissions as,
$$E(b)_s^f = E_s^f \, F(b)_s^f \tag{4}$$
where $E$, is the FFCO$_2$ emissions either for the county or for building, $b$, and fuel. In allocating emissions from coal
consumptions, however, $NE\text{-}EUI$ takes the value of "1" for all building types so that the allocated emission in a
building is directly proportional to the floor area.
**2.3.1.2 Temporal distribution**
The hourly time structure for buildings in the residential and commercial sectors are created via the use of eQUEST,
a building energy simulation tool run for each of the building classes listed in Table 2 and Table 3 and using only
the temporal structure of the energy consumption output (*Hirsch & Associates*, 2004). The model domain is
specified as the city of Los Angeles for the year 2011 with TMY weather data from the DOE (*Marion and Urban*,
1995). The mean building area is provided by the parcel data as described previously.
For the industrial buildings, a temporal profile representing the mean of industrial point source temporal surrogates
provided by EPA, are used (USEPA, 2015a). Figure 3 shows the hourly time profile during a one-week period in
April for a selected building in the residential and commercial sector, respectively.



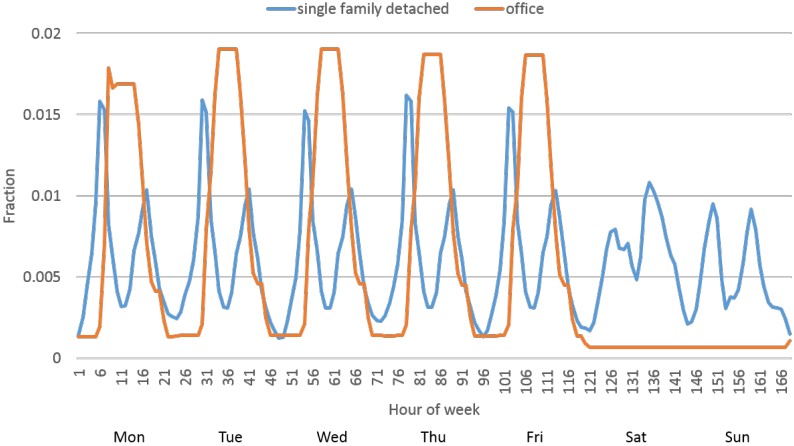


**Figure 3. Energy consumption intensity (hourly fraction) from an eQUEST simulation on the average week in 2011 for two types of buildings: "single family detached house" and "office".**

**2.3.2    Industrial and commercial point sources**
Little space/time processing is required for industrial and commercial point source emissions since they are
geocoded to specific facilities/emitting stacks or similar identifiable emission points. However, visual inspection of
the point source locations in GIS suggested potential geocoding errors. Point source locations were reviewed by
searching facility names to an online address search or via the EPA's Facility Registry Service (FRS) which can link
the facility in question to all the reporting made to the federal government under other environmental regulations
(USEPA, 2013). This often returns a more accurate physical location. The geolocations considered inaccurate were
manually corrected. Out of the total 192 facilities with corrected locations, 13 were moved a distance of between
924 and 1022 km while the remaining 179 were moved 0.5 km or less. The large magnitude location changes were
likely transcription errors when originally recording the location coordinates.
A given commercial or industrial point source is typically composed of multiple emission processes or units. For
example, in Los Angeles County, the 2011 NEI reports a total of 3409 emission records at 842 individual facilities.
In some cases, the multiple emitting points at a facility are not at exactly the same geocoded point but may represent
different emitting points at a facility that occupies a large area of land. Most often, however, all emitting points at a
given facility are geocoded to the same latitude and longitude.
The sub-annual temporal distribution for the commercial and industrial point source emissions used temporal
surrogate profiles provided by the EPA, linked according to the SCC of the emission record (USEPA, 2015a).
**2.3.3    Electricity production**
As described in Section 2.2, three different data sources are used to quantify the FFCO$_2$ emissions in the Hestia-LA
domain: the Clean Air Markets Division (CAMD), the DOE-EIA reporting and 2011 NEI CO emissions data. In
2011 there were a total of 34 CAMD facilities, 228 EIA facilities and 147 NEI facilities (reported through the NEI
2011 point source fileset) in the Hestia-LA domain. Total electricity production emissions in the domain was 6.21



MtC/year exclusive of biogenic fuels and 6.68 MtC/year with biogenics included. The CAMD data is reported at
hourly resolution, while the DOE EIA data is reported at monthly resolution and the 2011 NEI data is reported at
annual resolution only. Reduction of all data to an hourly time increment was achieved by maintaining constant
emissions within a month or year for the DOE EIA and 2011 NEI data, respectively.
**2.3.4    Onroad**
A preliminary version of the Hestia-LA onroad emissions estimates were presented by Rao et al. (2017). The version
presented here uses updated data and Hestia methodologies.
**2.3.4.1    Temporal distribution**
The Hestia-LA onroad $FFCO_2$ emissions input are retrieved from the Vulcan v3.0 output spatialized to specific road
segments in the Hestia-LA domain and categorized by vehicle class/fuel. Hence, no further spatialization was
required.
Construction of the temporal distribution in the Hestia system relies upon the California Department of
Transportation (CalTrans) Performance Measurement System (PeMS) (PeMS, 2018). This dataset contains 2011
traffic count data collected at 5 min intervals at measuring stations along freeways and principal arterials and along
some minor arterials and collectors (major and minor). Aggregation of the 5-minute counts to hourly values are used
to construct hourly fractions for each measurement station.
To apply a time distribution for the $FFCO_2$ onroad emissions on each road segment, an Inverse Distance Weighting
(IDW) spatial interpolation method was used. A search within a neighborhood of a 10 km radius is performed from
the midpoint of each road segment to locate PeMS sites using a nearest neighbor searching library (Mount and Arya,
2010). In cases where more than one station was available, the IDW interpolation was applied; in cases where only
one station was available, the time structure of this station was directly assigned to the road segment in question. In
cases where no station was available within the 10-km neighborhood, an average temporal distribution was assigned
(an average of all station values in a county at that hour for that road type). This last case occurred mostly in the
rural portions of predominantly rural counties.
For local roads, PeMS data was not available in any of the counties within the Hestia-LA domain. Instead, the
weekday hourly time fractions were generated from Annual Average Weekday Traffic (AAWT) data supplied by
SCAG (*Mike Ainsworth*, 2014). The data contained five distinct time periods within a single 24 hour cycle: 6-9 am,
9 am-3 pm, 3-7 pm, 7-9 pm, 9 pm-6 am. Hourly time fractions for weekends were derived from the county average
of weekend hourly time fractions. The weekday and weekend hourly time fractions were combined to form a
complete week, and then replicated for all 52 weeks in the entire year. This was done because there was no
significant seasonality in weekday and weekend traffic across the year as observed from PeMS data.
**2.3.5    Nonroad**
The nonroad Hestia-LA $FFCO_2$ emissions are completely determined in the Vulcan system and hence, passed to the
Hestia-LA domain without further processing (see Gurney et al., 2018 for details). To summarize the Vulcan

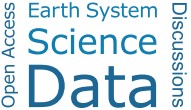

process, California did not report FFCO$_2$ nonroad emissions to the NEI 2011 but did report nonroad CO emissions.
The CO emissions were converted to FFCO$_2$ using the SCC-specific ratios of CO$_2$/CO derived from all other states
that reported both species (a mean value). The spatial distribution of the nonroad FFCO$_2$ emissions followed two
approaches. Nonroad FFCO$_2$ emissions reported through the 2011 NEI point data source (5 locations, 12% of
nonroad FFCO$_2$ in the LA Megacity) are located in space according to the provided latitude and longitude.
Emissions reported through the county-scale nonroad data source utilize multiple spatial surrogates provided by the
EPA reflecting a series of spatial entities such as the mines, golf courses and agricultural lands. There were instances
in which nonroad FFCO$_2$ emissions could not be associated with a spatial entity due to missing data. These
emissions are spatialized by first aggregating all the offending sub-county emission elements within a county for a
given surrogate shape type (e.g., golf courses, mines) and then distributing these emissions evenly across the county.
To distribute the nonroad FFCO$_2$ emissions from the annual to hourly timescale, a series of surrogate time profiles
provided by the EPA are used. These temporal surrogates are comprised of three cyclic time profiles (diurnal,
weekly, monthly) specific to SCC that are combined to generate hourly SCC-specific time fractions for an entire
calendar year.
**2.3.6 Airport**
Emissions of FFCO$_2$ from airports retrieved from the Vulcan system for the Hestia-LA domain are specific to
geocoded airport locations. Hence, the Hestia-LA system performs the temporal distribution only. There are 374
commercial airports/helipads in the Hestia-LA domain totaling 0.77 MtC/year, dominated by Los Angeles County
(0.39 MtC/year), and LAX in particular.
The annual airport FFCO$_2$ emissions are distributed in time utilizing airport-specific flight volume data from four
datasets:
1) The Operations Network (OPSNET) data from the Federal Aviation Administration (FAA) which reports total
date-specific, daily flight volume (365 values) at specific airports for specific aircraft classes (FAA, 2018a);
2) "AIRNAV" data which reports average daily percentage flight volume for aircraft class at US airports and
facilities (Airnav.com, 2018);
3) The Enhanced Traffic Management System Counts (ETMSC) daily flight volume data from the FAA was for two
airports in the Hestia-LA domain (NTD and RIV) with mostly military operations (FAA, 2018b);
4) The Los Angeles World Airports (LAWA) data which reports hourly flight volume for Los Angeles International
airport (LAX), Ontario airport (ONT), and Van Nuys airport (VNY) (LAWA, 2014).
For three large airports (LAX, ONT, VNY), the daily aircraft class-specific flight volume (from OPSNET) and the
hourly data on flight volume (from LAWA) were combined to create hourly aircraft class-specific time profiles
(Figure 4-6). All of the flight volume data are specific to four aircraft classes: Military (MIL), Air Carrier (AC),
General Aviation (GA), and Air Taxi (AT).



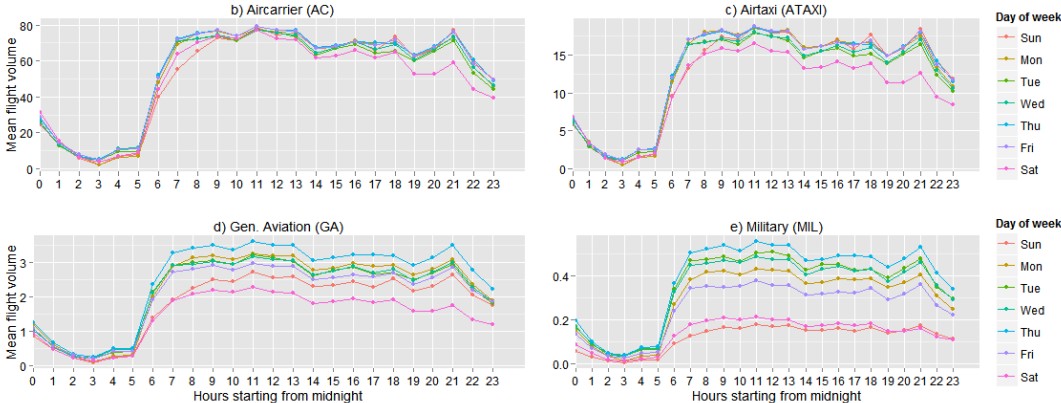

**Figure 4. Average hourly flight volume at LAX for a) total, b) AC, c) AT, d) GA, and e) MIL aircraft classes**
**for each day of the week. The plots represent the mean diurnal cycle for all Mondays, Tuesday, Wednesdays,**
**and so on, given a full year of data.**

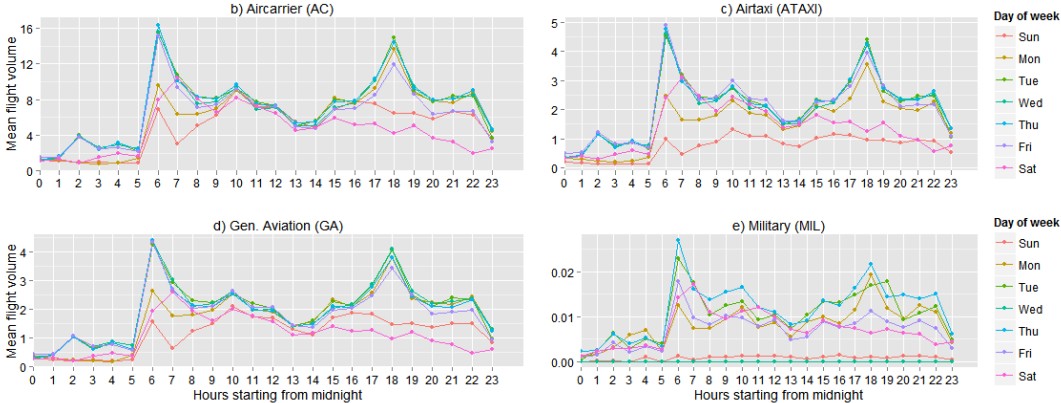


**Figure 5. Same as figure 4 but for the Ontario (ONT) airport.**

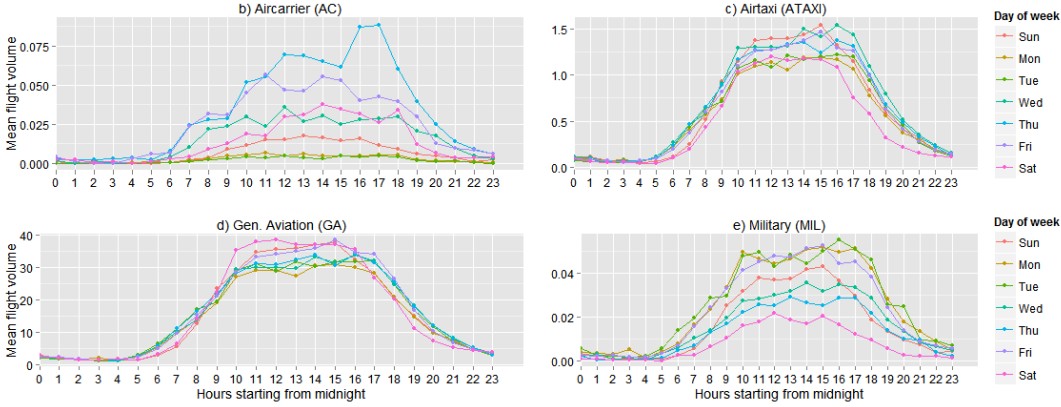


**Figure 6. Same as figure 4 but for the Van Nuys (VNY) airport.**



To generate hourly time profiles for all other airports in the Hestia-LA domain for which this type of detailed hourly
data was not available, airports first were categorized based on average daily flight volumes and average aircraft
class proportions from the OPSNET, AIRNAV and ETMSC data. Each airport was categorically matched to one of
the two non-international airports with hourly data (ONT, VNY) and the hourly time fractions adopted. LAX was
unique in terms of its volume and aircraft class proportions and hence was not used for any other airports. For
helipads and very small airports, a flat time structure was used.
**2.3.7 Railroad**
Railroad FFCO$_2$ emissions are similarly distributed in space within the Vulcan system and passed through to the
Hestia-LA landscape without alteration (see Gurney et al., 2018 for additional details). The Vulcan process treats
railroad point records somewhat differently from the railroad nonpoint records. The point source railroad emissions
are associated with rail yards and related geo-specific locales and are placed in space according to the provided
latitude and longitude. The railroad FFCO$_2$ emissions associated with the nonpoint 2011 NEI reporting contain an
ID variable that links to a spatial feature (rail line segment) in the EPA railroad GIS Shapefile. Nearly two-thirds of
the railroad emitting segments have no segment link. The sum of these "unlinked" railroad FFCO$_2$ emissions are
distributed to rail line within the given county according to freight statistics. The annual railroad FFCO$_2$ emissions
are distributed to the hourly timescale with no additional temporal structure (a "flat" time distribution).
**2.3.8 Commercial marine vessels**
The commercial marine vessel (CMV) FFCO$_2$ emissions retrieved from the Vulcan system are specific to county
and SCCs which are subsequently aggregated by the Hestia-LA system into emissions associated with two activity
categories: "port" emissions "underway". For the port CMV emissions (Figure 7), a port Shapefile from the EPA
was used as a reference along with a visual inspection of the coastline (USEPA, 2015a).

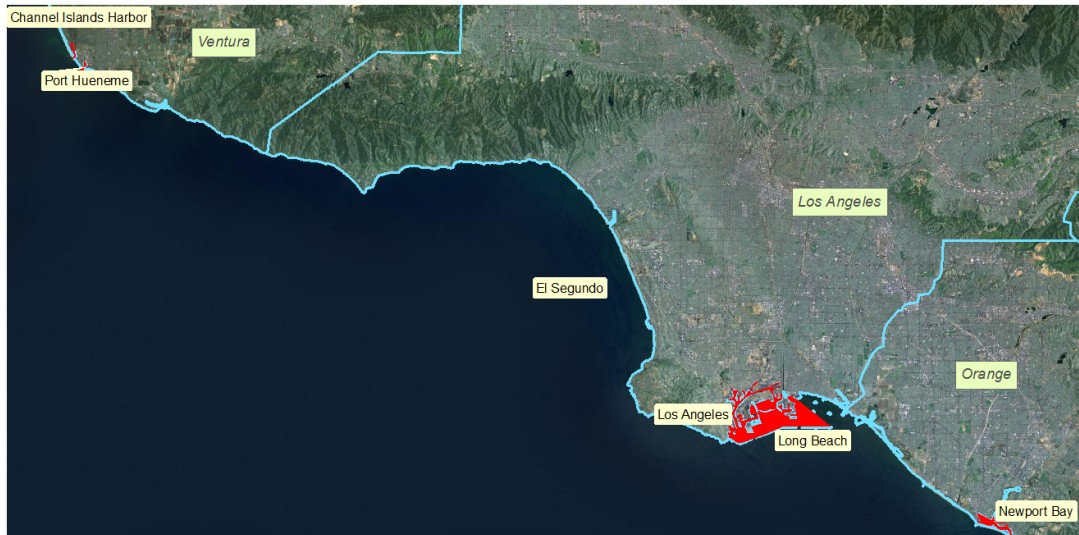

**Figure 7. The 6 ports in the Hestia-LA domain to which Vulcan FFCO$_2$ port emissions are allocated.**



Allocation of the FFCO$_2$ emissions designated as "underway" used a polyline Shapefile (Figure 8) of commercial
shipping lanes in California provided by CARB (Alexis, 2011). The shipping lanes for each county were bounded so
that only lanes between the exterior of ports and a distance of 24 miles from the port exterior, were included. County
total FFCO$_2$ emissions were then distributed evenly to these shipping lanes on a per unit length basis individually for
each of the three counties. Each shipping lane segment receives its length fraction of the annual total of underway
emissions.

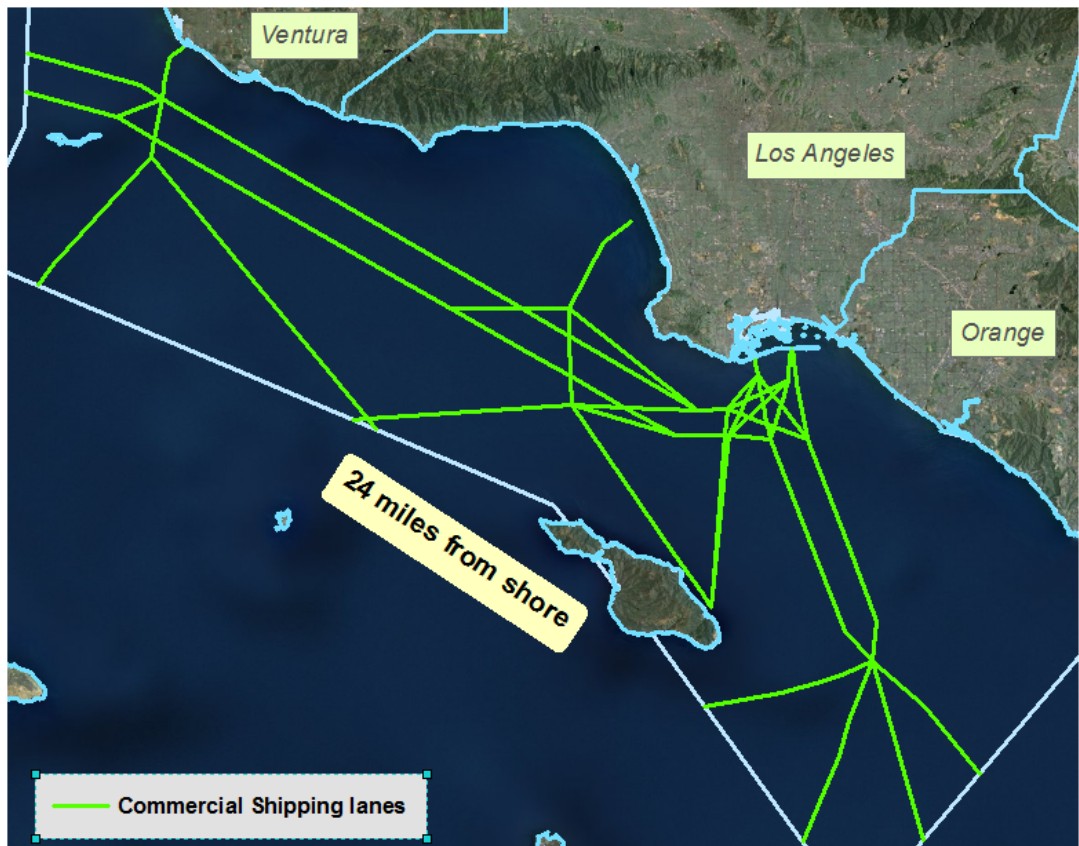

**Figure 8. Commercial Marine Vessel (CMV) shipping lanes in the Hestia-LA to which Vulcan FFCO$_2$**
**underway emissions are allocated.**
The time profile was based on the Marine Emissions Model (MEM) developed by CARB. MEM had marine vessel
activity data which includes the arrival time of ocean-going vessels for all ports in California spanning the 2004 to
2006 time period (Alexis, 2011). This hourly dataset was analyzed using a Fourier time series which allowed for an
isolation of the dominant cycles of ship traffic in the data. Results from the Fourier fit were then used to fill in the
missing hours. Weekday hours were examined separately from weekend hours to isolate potential differences in
traffic volume. Three cycles resulted: a 24-hour diurnal cycle, a weekly cycle and a monthly cycle. These were
applied to all years of the annual FFCO$_2$ emissions to create an hourly distribution at each of the CMV ports within
the domain.



### 2.3.9 Cement

Emissions of FFCO₂ from cement production facilities retrieved from the Vulcan system for the Hestia-LA domain
are specific to geocoded facility locations. $CO_2$ is emitted from cement manufacturing as a result of fuel combustion
and as process-derived emissions [van Oss, 2005]. The emissions from fuel combustion are captured in the industrial
sector. The process-derived $CO_2$ emissions result from the chemical process that converts limestone to calcium
oxide and $CO_2$ during "clinker" production (clinker is the raw material for cement which is producing by grinding
the clinker material). These emissions are reported as cement sector emissions
These emissions are fully calculated, spatialized and temporalized in the Vulcan v3.0 system and passed directly to
the Hestia-LA landscape. The cement facilities are geocoded with some corrections to provide more accurate
placement of the emission stacks.

### 2.4 Gridding

The county-level FFCO₂ emissions inventory, which has been distributed into the point, line and polygon features
by sector, are rasterized into a sector-specific and time-resolved gridded form under a common grid reference. This
grid reference divides the entire Hestia-LA domain into 509-by-342 1 km x 1 km grid cells on the California State
Plane Coordinate System. The grid reference is made into "fishnet" in the Shapefile format with 509-by-342 square
geometries.
The first step of the gridding procedure is to perform a spatial intersection operation between the fishnet and each of
the sectoral emissions layers in ArcGIS. The output of an intersection operation is a new set of features common to
both input layers. The emissions value of each feature in the intersection output was scaled by the ratio of the spatial
footprint of the feature to that of the original feature in the sectoral emissions layer. For line-source and polygon-
source emissions layers, the spatial footprint represents the line length and polygon area respectively. For point-
source layers, the footprint is equal to 1.

### 2.5 Uncertainty

Uncertainty estimation for Hestia results are challenging owing to the fact that many of the datasets used to
construct the flux results are not accompanied by uncertainty or traceable to transparent sources or methods. The
approach taken for the Hestia-LA v2.5 results was to conservatively estimate the uncertainty based on available
comparisons to Hestia results and exploration of the dominant components of the Hestia output. The first of these is
a comparison of the Hestia-Indianapolis (Hestia-Indy) results to an inverse-estimation of fluxes in the INFLUX
project (Gurney et al., 2017). In that study, it was shown that the Hestia-Indy whole-city FFCO₂ emissions result
agreed with an inverse estimate (Lauvaux et al., 2016) within 3.3% (CI: -4.6% to +10.7%). This suggests both
potential bias (3.3%) and an estimation uncertainty (~7.5%). This comparison was accomplished by estimating
portions of the carbon budget, included in the inverse estimate, but not explicitly included in the Hestia-Indy result.
Most importantly, biosphere respiration estimated from chamber studies at commensurate urban latitudes combined
with a remote-sensing based approach to quantifying the available vegetated landscape. This comparison, it should
be noted, is for a single city (Indianapolis) for a single time period. We directly sum the random and systematic error
and use this in the current study to represent the Hestia-LA whole-city uncertainty (a 95% CI), rounded up to 11%.

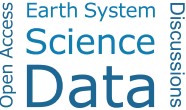

The next element for consideration with a conservative uncertainty estimate is the work done to compare two
different electricity production FFCO$_2$ estimates in the US. This work (Gurney et al., 2016) found that one-fifth of
the facilities had monthly FFCO$_2$ emission differences exceeding -6.4%/+6.8% for the year 2009 (the closest
analyzed year to the 2011 analysis examined here). The distributions of emissions of the two datasets were not
normally distributed nor were the differences. Hence, a typical gaussian uncertainty estimate cannot be made –
rather, the difference distribution was represented by quintiles of percentage difference. Hence, these values cannot
be cast within the context of other normally-distributed errors. However, we conservatively consider the quintile
value (the positive and negative tails) as a one-sigma value and 13% as a two-sigma value. The contribution of
electricity production is important to urban FFCO$_2$ emissions uncertainty given how large power production can be
within the total urban FFCO$_2$ context. For example, in the Los Angeles Megacity electricity production accounts for
19% of the total FFCO$_2$ emissions. The percentage differences can act as a form of uncertainty at the pointwise or
(conservatively) the gridcell scale, though only representative of the type of uncertainties represented by electricity
production point sources.
Finally, an initial assessment of the range of two critical parameters in the Vulcan/Hestia estimation is included as
part of the conservative uncertainty estimation. The two critical parameters are the CO emissions factor and the CO$_2$
emissions factor. Primarily for the CO EF, there is a range of potential values for each application (combination of
fuel category and combustion technology) though that range is not represented by a well-populated distribution of
values, but rather a discrete set of values within the data sources described in Gurney et al. (2009). Furthermore, the
expectation is that the CO EFs would not be normally distributed even were there to be a well-populated distribution
of values (i.e. many literature estimates of the same fuel/combustion technology) owing to the nature of CO
emissions from fuel combustion. This is driven by both the variation in combustion conditions for a given
fuel/technology combination and the variation is CO EF values across combustion technology. The distribution
would likely be a positively skewed "heavy" or "long" tailed distribution. For the current study, a range of the CO
and CO$_2$ EF values culled from the literature are conservatively assigned a one-sigma uncertainty of 10% or a two-
sigma value of 20%. Like the electricity production analysis in the previous paragraph, the uncertainty associated
with the CO and CO$_2$ emission factors is a gridcell-scale uncertainty (as opposed to whole city where error
cancelation occurs) and is independent of the electricity production uncertainty estimate (the CO and CO$_2$ EF values
are not used in the electriity production sector but in the other point sources and nonpoint sources).
These latter two uncertainty are more representative of gridcell-scale uncertainties and sum them in quadrature to
arrive at a gridcell-scale uncertainty (95% CI) of 23.4% or conservatively rounded to 25%. Work is underway that
includes a complete input parameter range for the Hestia emissions data results to more formally assign uncertainty
at multiple scales.
**3 Results**
The total 2011 emissions for the Hestia-LA domain are 48.06 ± 5.3 MtC/yr (Figure 9, Table 5). Transportation
accounts for the largest share (24.27 ± 2.7 MtC/yr) of the total and within the transportation sector, onroad emissions
account for the largest portion (20.81 ± 2.3 MtC/yr). The next largest sectors are the industrial (11.65 MtC/yr ± 1.3)
and electricity production (5.88 ± 0.76 MtC/yr) sectors, respectively. Onroad, electricity production, residential and




industrial FFCO$_2$ emissions make up 86% of the total. Petroleum accounts for almost 75% of the total LA Megacity
fuel consumption for direct FFCO$_2$ emissions consistent with the dominance of the transportation and industrial
sectors which are mostly reliant on petroleum fuels. Los Angeles County dominates emissions in the five counties of
the Hestia-LA domain accounting for 55% of the total FFCO$_2$ emissions. This is followed by San Bernardino,
Orange, Riverside, and Ventura counties, respectively. Los Angeles and San Bernardino counties are dominated by
onroad and industrial FFCO$_2$ emissions, while onroad emissions account for the largest share, by far, in the
remaining three counties. Not surprisingly, Los Angeles county has the largest CMV FFCO$_2$ emissions among the
five counties owing to the port of Los Angeles which hosts a large amount of international commercial shipping. At
0.61 ± 0.067 MtC/yr, it rivals in emission magnitude the combination of residential and commercial building
emissions in each of the other four counties.

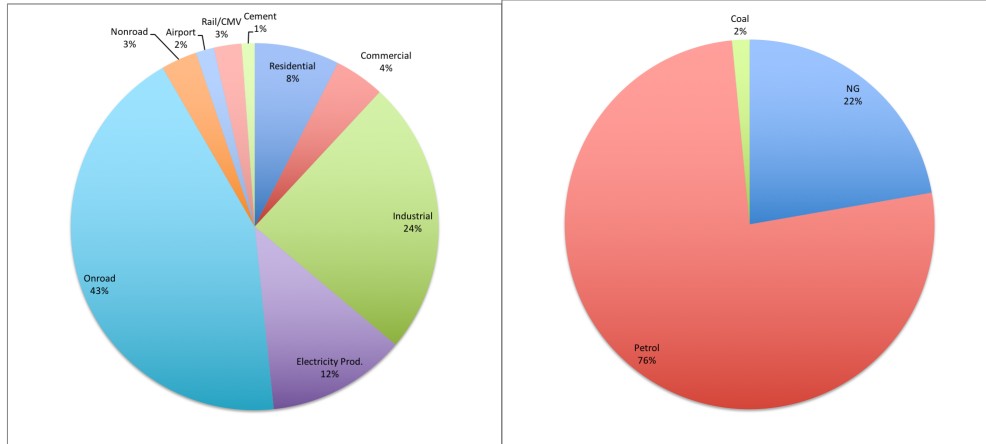


**Figure 9. Total FFCO$_2$ emissions proportions for the Hestia-LA domain. a) FFCO$_2$ emission proportions by**
**sector; b) FFCO$_2$ emission proportions by fuel category.**
**Table 5. Sectoral FFCO$_2$ emissions in the five Hestia-LA domain counties for the year 2011. Units: MtC/yr.**

| Sector | Los Angeles (MtC/yr) | Orange (MtC/yr) | San Bernardino (MtC/yr) | Riverside (MtC/yr) | Ventura (MtC/yr) | Total (MtC/yr) |
|---|---|---|---|---|---|---|
| Residential | 2.00 | 0.64 | 0.40 | 0.36 | 0.20 | 3.59 |
| Commercial | 1.47 | 0.12 | 0.21 | 0.24 | 0.071 | 2.12 |
| Industrial | 7.27 | 0.94 | 2.99 | 0.25 | 0.20 | 11.65 |
| Electricity production | 2.73 | 0.69 | 1.54 | 0.71 | 0.21 | 5.88 |
| Transportation | 12.95 | 3.83 | 3.58 | 2.88 | 1.02 | 24.27 |
|    Onroad | 11.03 | 3.46 | 2.98 | 2.51 | 0.82 | 20.81 |
|    Nonroad | 0.79 | 0.27 | 0.19 | 0.19 | 0.087 | 1.52 |
|    Airport | 0.39 | 0.06 | 0.14 | 0.11 | 0.070 | 0.77 |
|    Railroad | 0.13 | 0.028 | 0.27 | 0.072 | 0.010 | 0.51 |
|    CMV | 0.61 | 0.012 | 0 | 0 | 0.037 | 0.66 |
| Cement | 0 | 0 | 0.55 | 0.0077 | 0 | 0.55 |
| **Total** | **26.42** | **6.22** | **9.28** | **4.45** | **1.70** | **48.06** |

Total emissions in the LA Megacity show a small downward trend over the 2010-2015 time period of 0.44%/year
which is a statistically significant trend (slope: -0.21 MtC/yr; CI: -0.397, -0.023). Individual sectors show greater
variation there are compensating temporal changes among the individual sectors (Figure 10). The residential sector
showed a relatively large decline in 2014, though due to its relatively small portion of total emissions, has limited



impact on the total temporal variation from 2010-2015. Similarly, 2015 showed a large increase in commercial
sector emissions which also do not translate to large changes in the total $FFCO_2$ emissions time series. The relative
temporal stability of the industrial and onroad $FFCO_2$ emissions sectors combined with their large share of the total
$FFCO_2$ emissions are reflected in the total emissions trend. When categorized by fuel type, natural gas $FFCO_2$
emissions exhibited the greatest variation with a maxima in 2012 and to a lesser extent 2013, driven primarily by
consumption in the electricity production sector.

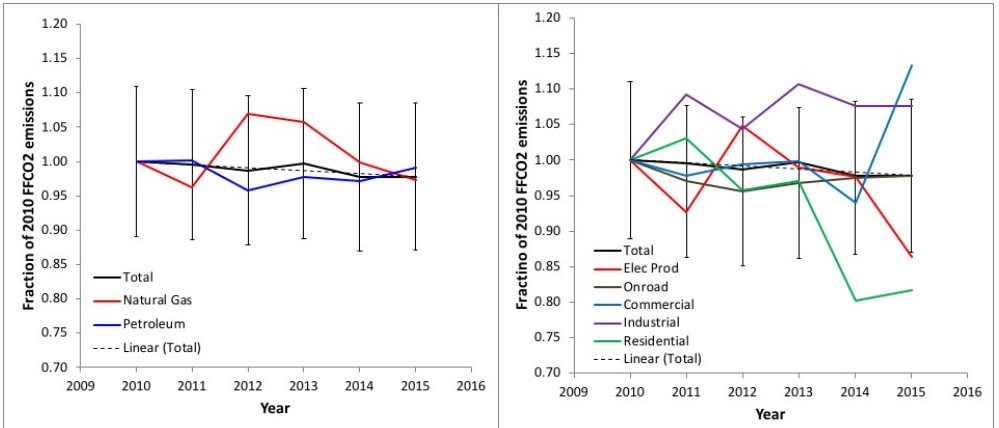


**Figure 10. Fractional changes over the 2010 to 2015 timeframe in LA Basin $FFCO_2$ emissions. a) by fuel
category; b) by sector. Whole-city error provided for the total $FFCO_2$ emissions only.**

Spatial distribution of the Hestia-LA $FFCO_2$ emissions demonstrate the importance of the populated areas and road-
intensive portions of the domain in the overall emissions (Figure 11). The constant emissions that appear over large
areas, particularly in San Bernardino and Riverside counties, are due to the nonroad $FFCO_2$ emissions which have
relatively simple spatial distribution proxies with considerable areal extent.



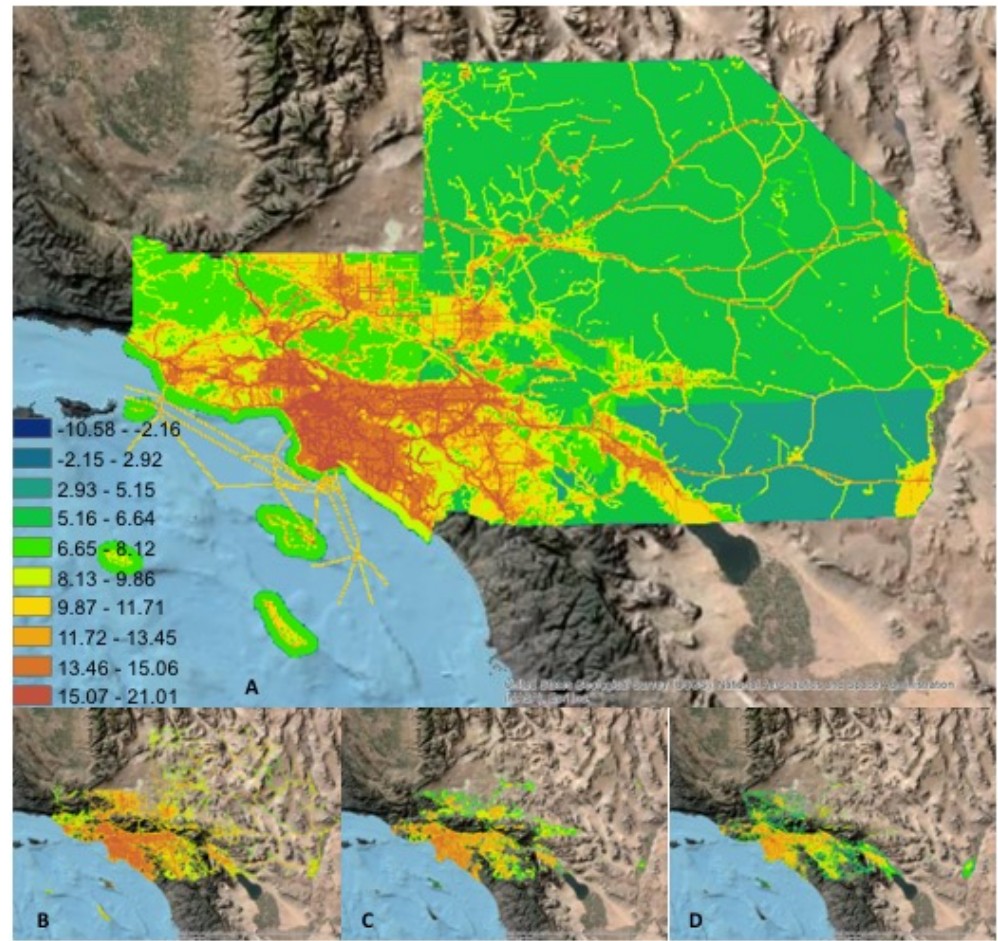

**Figure 11. Hestia-LA v2.5 FFCO$_2$ emissions for the year 2011 represented on a 1 km x 1 km grid. a) total FFCO$_2$ emissions; b) onroad FFCO$_2$ emissions; c) residential FFCO$_2$ emissions; d) commercial FFCO$_2$ emissions. Units: natural logarithm KgC/gridcell/yr.**

Figure 12 shows the cumulative FFCO$_2$ emissions across four of the sectors for which the 1 km$^2$ gridcell accumulation is most appropriate: the commercial, industrial, onroad, and residential sectors. The other FFCO$_2$ emission sectors (airport, electricity production, cement) are not included in Figure 12 because they are dominated by a few points, have limited spatial distribution (railroad) or no spatial variance (nonroad). The accumulation of FFCO$_2$ emissions at the threshold by which 10% of the gridcells are accumulated is noted on the figure. For the industrial sector, 10% of the largest emitting gridcells account for 93.6% of the total industrial sector emissions. For the commercial sector this occurs at 73.4% of the accumulated gridcells. For the onroad and residential sectors this occurs at 66.2% and 45.3%, respectively. This demonstrates two important points about the FFCO$_2$ emissions in the Los Angeles Megacity (and most cities). First, the emissions have very high spatial variance with few gridcells accounting for a large portion of the total FFCO$_2$ emissions. Second, this is particularly true for the industrial sector, driven by the fact that it is comprised of a large proportion of point emitters. This is somewhat true of the

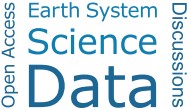

commercial sector which does have some pointwise data within the original NEI reporting. Of the remaining two
sectors, which contain no pointwise spatial emitters, the majority (66.2%) of the onroad emissions are captured in
the largest 10% while the residential sector, being less concentrated, shows an accumulation just short of the 50%
threshold at a 10% gridcell accumulation threshold.

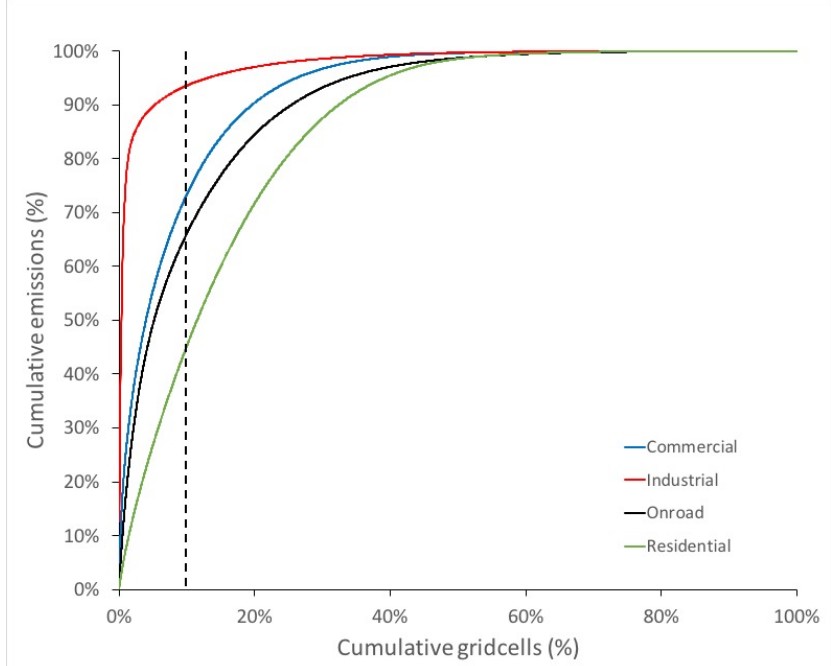


**Figure 12. Cumulative FFCO$_2$ emissions according to key sectors in the Hestia-LA FFCO$_2$ emissions data
product. The dashed line at 10% cumulative grid cells is given for reference. See text for details.**

An important attribute of estimating urban emissions at fine space and time scales is the resulting clustering in space
(and time) of the emissions and the varying patterns of the clustering across the emitting sectors. Figure 13 provides
an analysis of spatial clustering using the *Getis-Ord-Gi* statistic which provides a score that measures statistically
significant departures from random local clustering (*Getis and Ord*, 1992). The three sectors included in this
analysis are the residential, commercial and onroad sectors. The onroad sector shows a more widely dispersed
clustering pattern with local "hotspots" generated by high traffic flow points and traffic congestion, primarily on the
interstate network coincident with a greater density of commercial and residential activity. The residential sector
exhibits less extensivity compared to the onroad FFCO$_2$ emissions clustering but with larger individual hotspot
areas. Particularly large clustering occurs from the coast centered on Santa Monica and Marina del Rey and
extending East and North through West Hollywood on to Pasadena and Alhambra. Other hotspots occur in the
Manhattan Beach to Redondo Beach corridor, the Burbank and Glendale area and the coastal portion of Orange
county (e.g. Huntington Beach, Newport Beach). The commercial sector shows the a similar overall extensivity to
the residential sector but with less extensive individual hotspots associated with commercial building clusters.

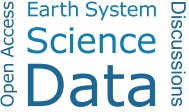




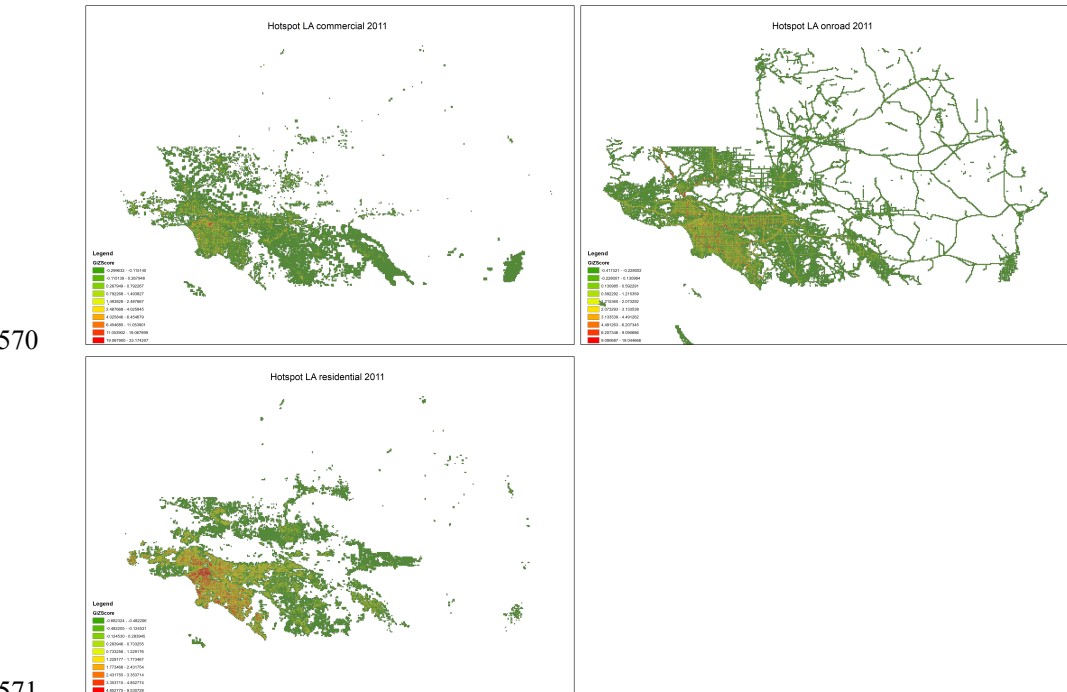

**Figure 13. The *Getis-Gi* z-score for Hestia-LA FFCO₂ emissions across three sectors; a) commercial; b)**
**onroad; c) residential.**
There are very few potential sources for comparison to the Hestia FFCO₂ emissions as few inventory efforts have
been accomplished at the sub-state spatial scale in the United States. However, the Southern California Association
of Governments (SCAG) have completed a regional greenhouse gas emissions inventory for a base year period of
1990-2009 with projections out to the year 2035 (SCAG, 2012). The SCAG inventory reflects two components that
make comparison to the Hestia-LA FFCO₂ emissions data product imperfect. First, the domain considered in the
SCAG inventory includes Imperial county, a county not included in the Hestia-LA domain. However, Imperial
county is estimated to be less than a few percent of the SCAG domain total. For example, Imperial county onroad
VMT is 1.9% of the SCAG domain total. The Imperial county retail sales of electricity is 1.1% of the SCAG domain
total. The other distinction is that the SCAG inventory reports total GHGs, inclusive of both methane (CH₄) and
nitrous oxide (N₂O). However, in the sectors and activities used in comparing the SCAG inventory to the Hestia-LA
FFCO₂ emissions data product, both CH₄ and N₂O are negligibly small. Hence, small differences (<5%) could be
due to these categorical discrepancies.
Figure 14 shows a 2010 comparison between the two estimates using the comparable sector divisions. The Hestia-
LA FFCO₂ emissions estimate is 10.7% larger than the SCAG estimate, 95% of the difference (4.46 MtC/yr) owing
to the larger industrial and electricity production FFCO₂ emissions in the Hestia estimate. We have included the
nonroad sector in the onroad category as the SCAG inventory did not explicitly include a nonroad sector. SCAG
documentation suggests that the nonroad sector is included in the forecasts for the residential, commercial and
industrial sectors (SCAG, 2012, page C-10) but further details on the base year estimates could not be found and no

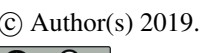



mention is made in the report where these sectors are described. If the Hestia nonroad estimate (1.56 MtC/yr) were
not allocated to onroad but distributed to the residential, commercial and industrial sectors it would exacerbate the
difference in the onroad, commercial and industrial sectors.

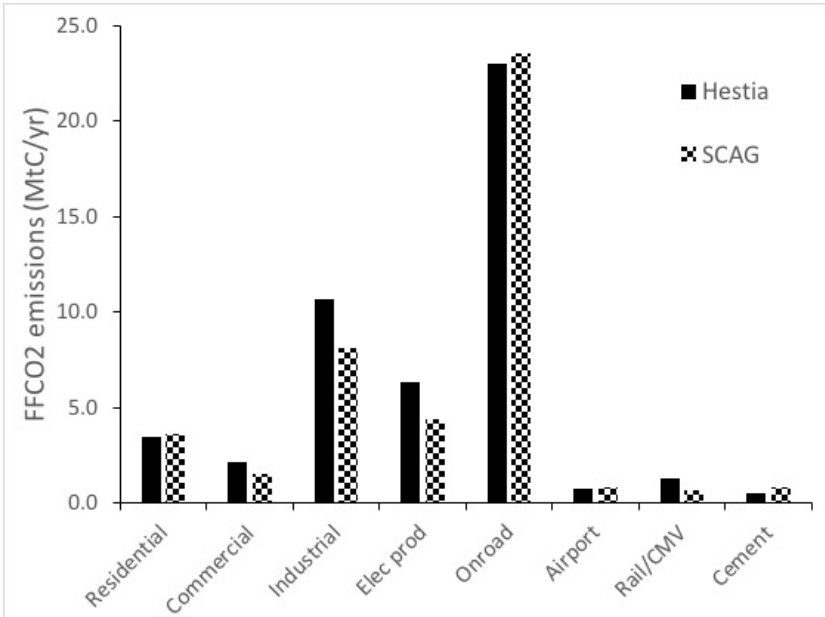


**Figure 14. Comparison of sector-specific FFCO₂ emissions for the year 2010 between the Hestia-LA and**
**SCAG estimates. Units: MtC/yr.**
The California Energy Commission archives energy consumption data for both natural gas and electricity
(http://ecdms.energy.ca.gov/). The data is archived as specific to the residential sector and the non-residential sector.
Because of ambiguities regarding the non-residential sector definition, we compare the reported values by county for
the residential only (Table 6). Good agreement for natural gas FFCO₂ emissions is achieved for the Los Angeles
Megacity as a whole (<1%) with some variation at the scale of the individual counties. Agreement with the CEC
estimate is better than that found for the comparison with the SCAG inventory (Hestia being 3.1% lower than the
SCAG residential NG FFCO₂ estimate).
**Table 6. Residential natural gas FFCO₂ emissions in the five Hestia-LA domain counties for the year 2011**
**compared to estimates from the California Energy Commission (CEC). Units: MtC/yr.**

| County | Hestia | CEC | diff (%) |
|---|---|---|---|
| Los Angeles | 1.94 | 1.98 | -2.0% |
| Orange | 0.63 | 0.59 | 5.7% |
| San Bernardino | 0.40 | 0.39 | 0.8% |
| Riverside | 0.35 | 0.39 | -11.1% |
| Ventura | 0.19 | 0.18 | 6.5% |
| **LA Megacity** | **3.51** | **3.54** | **-0.9%** |

Average hourly variations in FFCO₂ emissions are sensitive to both the sector and spatial location. Figure 15
presents annual mean diurnal patterns specified by county and sector (the railroad or cement sectors were
constructed with no diurnal cycle and hence is not shown). As noted previously, Los Angeles county shows the



greatest emissions overall, particularly for the commercial marine vessel sector where the port of Los Angeles
dominates. The commercial, residential, onroad and CMV sectors exhibit two maxima, one in the morning (~5-10
am, local time) and another in the afternoon/evening. In the commercial sector, this afternoon/evening maximum
occurs later in this time period centered on 9 pm local time, coinciding with retail closing schedules. The maximum
CMV emissions are shifted by roughly two hours earlier in the day for both the morning and afternoon/evening
peaks. The afternoon/evening maximum for the onroad sector shows an afternoon/evening maximum that is of
longer duration than that in the morning with emissions gradually rising after the midpoint of the day, local time. In
addition to large daily variations, the onroad sector contains a significant weekly temporal pattern with emissions
largest on Monday and smallest on Saturday (Figure 16).
Diurnal patterns in onroad and airport $FFCO_2$ emissions have a single maximum at the middle of the day but broadly
extending across all daylit hours. In the case of the nonroad emissions, this is simply a reflection of the EPA
temporal surrogate applied. In the case of the airport $FFCO_2$ emissions, the time structure reflects the reported air
traffic volume at the major airports in the LA Megacity. Finally, the industrial and electricity production sectors
maintain relatively constant emissions across all 24 hours. In the case of the industrial sector, this reflects the
integration of industry-specific EPA temporal surrogates within a given county. For the electricity production sector,
the time structure is primarily driven by the stack-monitored emissions and shows a slightly greater emission in the
evening hours compared to all other hours.
The diurnal patterns are consistent across all five counties with the exception of the commercial sector where there
are small differences in the maximum point of the morning emissions in San Bernardino and Ventura counties
compared to the other LA Megacity counties.




a)                                      b)

c)                                      d)

e)                                      f)

g)                                      h)

**Figure 15. Average daily FFCO₂ emissions in the Hestia-LA v2.5 data product for five counties across eight**
**sectors. A) residential; b) onroad; c) commercial; d) airport; e) commercial marine vessel; f) electricity**
**production; g) industrial; h) nonroad. Note: different scale range on each plot. Units: kgC/hour.**

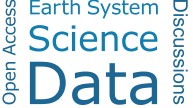

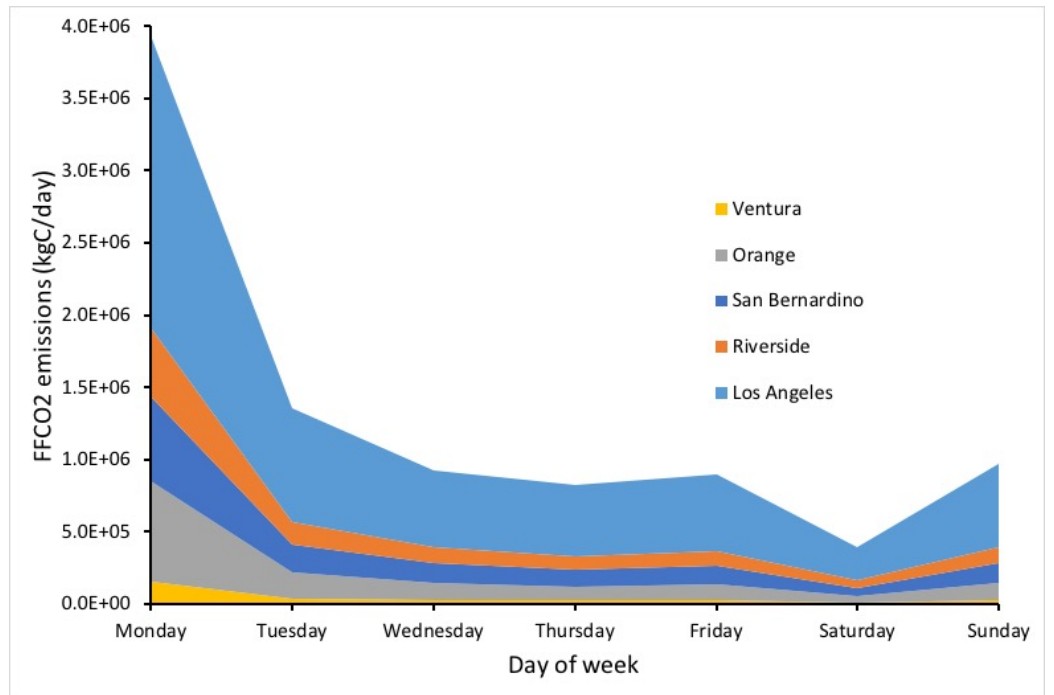


**Figure 16. Average weekly onroad FFCO₂ emissions from the Hestia-LA v2.5 data product for five counties.**
**Units: kgC/day**


**4 Discussion**
The first Hestia urban FFCO₂ emissions data product was produced for the Indianapolis domain (Gurney et al.,
2012). As an outcome of the Hestia effort, a large multifaceted effort, the Indianapolis Flux Experiment (INFLUX),
emerged (Whetstone et al., 2017; Davis et al., 2017). INFLUX aims to advance quantification and associated
uncertainties of urban CO₂ and CH₄ emissions by integrating a high-resolution bottom-up emission data product,
such as Hestia, with atmospheric concentration measurements (Turnbull et al., 2015; Miles et al., 2017; Richardson
et al., 2017), flux measurements (Cambaliza et al., 2014; 2015; Heimberger et al., 2017), and atmospheric inverse
modeling. In addition to its use as a key constraint in the INFLUX atmospheric inverse estimation (Lauvaux et al.,
2016), Hestia has been informed by atmospheric observations making it useable as a standalone high-resolution flux
estimate offering a detailed space-time understanding of urban emissions. Begun in the late 2000s, INFLUX has
explored many aspects of the individual elements of a scientifically-driven urban flux assessment (e.g. Wu et al.,
2018) in addition to demonstrating potential reconciliation between Hestia and the atmospheric measurements
(Gurney et al., 2017; Turnbull et al., 2018). Similar efforts are ongoing in the Salt Lake City (Mitchell et al., 2016;
Lin et al., 2018) and Baltimore (Martin et al., 2018) domains with a different arrangement of atmospheric
monitoring and modeling. As with INFLUX, a Hestia FFCO₂ emissions data product was produced in each domain
(Patarasuk et al., 2016; Gurney et al., 2018).

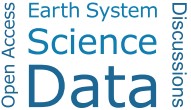

The Hestia Los Angeles Megacity effort was developed under the Megacities Carbon Project framework (https://megacities.jpl.nasa.gov/portal/). It was designed to serve the Megacities Carbon Project in a similar capacity to its role in INFLUX. The Hestia-LA results are unique in that it is the first high-resolution spatiotemporally-explicit inventory of $FFCO_2$ emissions centered over a megacity. Presented here at the 1 $km^2$ spatial and hourly temporal resolution, the emissions can be represented at finer spatial scales down to the individual building, though with higher uncertainty. While policy emphasis in California thus far has been focused on $CH_4$ emissions (Carranza et al., 2017; Wong et al., 2016; Verhulst et al., 2017; Hopkins et al., 2016), work is ongoing to use the extensive atmospheric $CO_2$ observing capacity in the Los Angeles domain (e.g. Newman et al., 2016; Wong et al., 2015; Wunch et al., 2009) within an atmospheric $CO_2$ inversion. This will offer an important evaluation of the Hestia-LA emissions for which limited independent evaluation is currently available.

The potential of the Hestia-LA $FFCO_2$ emissions to enable or assist with policymaking in the cities, counties or metropolitan planning domain of the overall Southern California area is considerable. The traditional urban inventory approach, such as accomplished by many cities as part of their climate action plans, are whole-city accounts, often specific to sector, that follow one of a few inventory protocols. Given the challenges of data acquisition and the idiosyncrasies of protocol choice and needs, the traditional urban inventories are difficult to compare across cities and hence, aggregate reliably in a metropolitan domain such as the LA Megacity. Importantly, without space and time explicit emissions information, they are difficult to calibrate with atmospheric measurements and hence, evaluate against this important scientific constraint. The Hestia-LA $FFCO_2$ emissions approach attempts to overcome these limitations to traditional inventory work. By quantifying emissions at the scale of individual buildings and road segments, with process detail such as the sector, fuel, and combustion technology, Hestia results can be organized according to most of the protocols in use by cities. This explicit space and time detail also allow for calibration to atmospheric measurements, for which emission location and time structure is essential.

The state of California continues to lead the nation in climate policy with numerous legislative and executive orders outlining both general reduction goals and specific policy instruments. The California Global Warming Solutions Act (Assembly Bill 32) passed in 2006, specifies a statewide reduction in greenhouse gas emissions to 1990 levels by the year 2020 (https://www.arb.ca.gov/cc/ab32/ab32.htm). Furthermore, the bill requires reporting and verification of reductions in order to demonstrate compliance. Executive order B-30-15 and Senate Bill, SB 32 have built on this with an aim to reduce emissions 40% below 1990 levels by 2030 and 80% below 1990 levels by 2050, respectively (https://www.gov.ca.gov/2015/04/29/news18938/; https://leginfo.legislature.ca.gov/faces/billTextClient.xhtml?bill_id=201520160SB32). Ultimately, much of the specific action needed to meet these goals will rest upon local governments and authorities. Given that 87% of the state population resides in urban areas and nearly half of state population resides in the Los Angeles Megacity, the cities and counties that comprise the Los Angeles metropolitan area have a central role to play in achieving the statewide climate change policy goals. The city of Los Angeles, the largest individual city in the metro region, has specified goals consistent with the state commitments, expecting to reduce greenhouse gas emissions 35% below 1990 levels by the year 2030 (http://environmentla.org/pdf/GreenLA_CAP_2007.pdf). To meet these reduction goals, policy actions will become increasingly difficult to achieve at no- or low-cost and economic efficiency will become central to making policy choices.



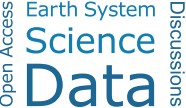

The most important attribute of the Hestia-LA approach, therefore, is the potential it offers for targeting urban $CO_2$
reduction policy more efficiently. As shown in Figures 12 and 13, $FFCO_2$ emissions are highly variable in space and
typically cluster in concentrated areas. In choosing specific policy approaches and instruments, this offers Los
Angeles policymakers the ability to target specific neighborhoods, road segments, or commercial hubs, where
policies will achieve the greatest reduction for resources expended. This rests on the argument that specificity leads
to efficiency. As all cities, including those in the Los Angeles Megacity, move towards those aspects of carbon
emission reductions that are not part of the "low hanging fruit" policy instruments, competition for limited resources
and policy justification will increase. Having information that targets the most efficient and effective emission
reduction investments, established by independent rigorous scientific information, will be at a premium. For
example, if a small proportion of the commercial sector buildings in the LA Megacity account for a large proportion
of the $FFCO_2$ emissions, knowing the location of these buildings and targeting energy efficiency programs to those
buildings, may offer the most economically efficient route to emissions reductions in the commercial sector. A
similar argument can be made in the onroad sector due to the clustering of large onroad emitting gridcells and
specific road-class attributes (see Rao et al., 2017).
A number of caveats are worth mentioning in association with the Hestia-LA v2.5 $FFCO_2$ emissions results. With
Vulcan v3.0 as the starting point for the quantification in Hestia, errors in Vulcan will be passed to Hestia, with a
few exceptions. Of particular note are the industrial sector and more specifically, refining operations which have
limited emissions reporting. These remain difficult to quantify due to the range of CO emission factors representing
many of the combustion processes undertaken at these large and complex facilities. The uncertainty estimation
described remains limited and there are additional sources of uncertainty that must be quantified such as categorical
errors (e.g. mis-specification of fuel category or road class), errors in spatial accuracy and spatial error correlation.
Quantifying these contributions to the overall uncertainty presented here remain a task for future work.
**5 Data availability, policy and future updates**
The Hestia-LA v2.5 emissions data product can be downloaded from the data repository at the National Institute
of Standards and Technology (https://doi.org/10.18434/T4/1502503) and is distributed under Creative Commons
Attribution 4.0 International (CC-BY 4.0, https://creativecommons.org/licenses/by/ 4.0/deed.en). The Hestia-LA
v2.5 $FFCO_2$ emissions data product is provided as annual and hourly (local and UTC versions) 1 km x 1 km
NetCDF file formats, one file for each of the 6 years (2010-2015). The hourly files are approximately 2.9 GB each.
The annual files are 0.34 GB each.
Attempts will be made to update the Hestia-LA $FFCO_2$ emissions on a roughly bi-annual basis, depending upon
support, the availability of updates to the Vulcan $FFCO_2$ emissions data product, and updates to the additional data
sources described in this study.
**6 Conclusion**
The Hestia Project quantifies urban fossil fuel $CO_2$ emissions at high space- and time-resolution with application to
both scientific and policy arenas. We present here the Hestia-LA version 2.5 $FFCO_2$ emissions data product which

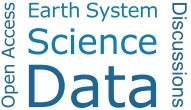

represents hourly, 1 km², sector-specific emissions for the five counties of the Los Angeles metropolitan area for the
2010 to 2015 time period. The methodology relies on the results of the Vulcan Project (version 3.0) further
enhancing and distributing emissions to the scale of individual buildings and road segments with local data sources
acquired from local government agencies. Each sector is quantified using data sources and spatial/temporal
distribution approaches distinct to the sector characteristics. The results offer a detailed view of $FFCO_2$ emissions
across the LA Megacity and point to the extreme spatial variance of emissions. For example, 10% of the 1 km²
emitting gridcells account for 93.6%, 73.4%, 66.2%, and 45.3% of the emissions in the industrial, onroad,
commercial, and residential sectors, respectively. We find that the LA Megacity emitted $48.06 \pm 5.3$ MtC/yr in the
year 2011, dominated by Los Angeles county ($26.42 \pm 2.9$ MtC/yr) and from a sector-specific viewpoint, dominated
by the onroad sector ($20.81 \pm 2.3$ MtC/yr). Hestia $FFCO_2$ emissions are 10.7% larger than the inventory estimate
generated by the local metropolitan planning agency, a difference that is driven by the industrial and electricity
production sectors. Good agreement is found (<1%) when comparing residential natural gas $FFCO_2$ emissions to
utility-based reporting at the county spatial scale. The largest temporal variations are found in the diurnal cycle with
the residential, commercial, onroad, and commercial marine vessel emissions showing to maxima, one in the
morning and a second in the afternoon/evening. Airport and nonroad emissions, by contrast show broad maxima
across the daylit hours. Finally, the industrial and electricity production sectors show little diurnal variation across
24 hours. The onroad sector also exhibits variation in the weekly distribution of emissions with maximum $FFCO_2$
emissions on Monday and minimum emissions on Saturday.
The Hestia-LA v2.5 $FFCO_2$ emissions data product offers the scientific and policymaking communities
unprecedented spatially and temporally-resolved information on $FFCO_2$ emission sources in the Los Angeles
Megacity. As part of the Megacities Carbon Project, future work includes incorporation into atmospheric $CO_2$
inversion research to further evaluate the Hestia-LA data product and improve estimation. Policymakers can use the
Hestia-LA results to better-understand $FFCO_2$ emissions at the human scale, offering the potential for improved
targeting of $FFCO_2$ reduction policy instruments. Finally, urban researchers can use Hestia-LA to explore a number
of important urban science questions such as how emissions intersect with other urban sociodemographic variables
such as income, education, housing size, or vehicle ownership.
The Hestia-LA data product is publicly available and will be updated with future years as data becomes available.
**Competing Interests.** The authors declare that they have no conflict of interest.
Acknowledgments. This research was made possible through support from the National Aeronautics and Space
Administration Carbon Monitoring System program, Understanding User Needs for Carbon Information project
(subcontract 1491755), the National Aeronautics and Space Administration grant NNX14AJ20G, the National
Institute of Standards and Technology grant 70NANB14H321 and 70NANB16H264, JPL's Strategic
University Research Partnership program, and the Trust for Public Land.



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
