# Peer review of "The Hestia Fossil Fuel CO2 Emissions Data Product for the Los 1 Angeles Megacity (Hestia-LA) 2"

_Earth System Science Data, 2018_

## Referee Comment (RC1) · Anonymous Referee #1 · 22 Mar 2019

This paper describes the development of a high space and time fossil fuel CO2 inventory for Los Angeles. It is an important contribution to the literature. The paper covers the description of the data set well. The one thing I would recommend is that the authors take a broader view of the community they are trying to influence with their ideas. The introduction is very self-referential and pays little to no attention to CO2 observing/modeling projects in Paris, Zurich, San Francisco, Boston and likely others. Helping the uninitiated reader to understand how this project relates to others should be a goal of the introduction and would make the paper more influential.

---

## Referee Comment (RC2) · Anonymous Referee #2 · 8 Apr 2019

General Comments

1. The Hestia data products provide bottom-up fossil fuel emissions at the urban scale with building/street and hourly space-time resolution. The data feed into atmospheric CO2 inversion studies and can help guide to climate change mitigation options, including disaggregation of national goals/policies to the local level. This paper focuses on the Los Angeles Megacity, the Combined Statistical Area that includes Los Angeles, Orange, Riverside, San Bernardino, and Ventura counties, providing CO2 emissions results at a spatial resolution of 1km by 1km for the years 2010-2015. The input data are from the Vulcan Project, adjusted where superior local and downscaled data are available. It was found that the study area emitted 48.06 MtC/yr. Of note, Hestia emissions were found to be 10.7% larger than the estimate by the local metropolitan

planning agency, the Southern California Association of Governments (SCAG).

2. The Hestia-LA data product, and the Vulcan dataset in general, are exciting advances in the urban greenhouse gas emissions quantification field. In particular, the data product goes beyond traditional carbon inventories used by city planning agencies to include a higher spatial and temporal resolution. This makes the data product particularly exciting for the development of highly impactful and targeted carbon reduction policies in cities.

Specific Comments

3. This reviewer has a keen interest in urban planning and policy-related efforts for greenhouse gas emissions reduction in cities. As such, the comments are intended to help improve the paper's accessibility to a planning/policy audience and help specify the policy-related importance of the Hestia-LA work.

4. This work is of interest to a variety of readers, including policy makers and urban planning professionals. With that in mind, I would recommend defining a few of the atmospheric-science-specific words and phrases earlier on in the introduction, particularly "flux" and "inversion", ideally in the context of other phrases that are more recognizable to the policy maker or layperson (e.g., "emissions", "CO2 emissions"). For example, the term "flux" begins to appear regularly on page 3 (e.g., "flux measurements", "flux estimation", "surface fluxes"), and it could be confusing to a reader who isn't used to the terminology. This would make the content more accessible to policy audiences.

5. The introduction does a good job in describing the body of work on greenhouse gas accounting in the urban environment, including the gap in existing methods that the Hestia Project aims to address. The discussion of policy-related issues on p2-3 are good: lines 50-55 (contributions from city-based policies to meeting national/global commitments), 64-68 (data and aggregation difficulties), 95-99 (translation to urban mitigation efforts). I would have liked to see a few more lines on the benefits and drawbacks of policy-related/traditional greenhouse gas emissions inventories (mentioned on p2, line 64 - note that these two citations do not actually appear in the reference list). It would be good to reference the Global Protocol for Community-scale GHG Emissions (GPC) here, as this is the current standard for city-based GHG inventories in the policy world, used by both the C40 and GCoM (two organizations that are currently mentioned in the paper). See https://ghgprotocol.org/greenhouse-gas-protocol-accounting-reporting-standard-cities. The paragraph on page 29, lines 672-683, does an excellent job of describing the issues with traditional urban inventories and the benefit of the Hestia approach. Could a few aspects of this description appear in the introduction?

6. The data collection and processing effort involved in creating the Vulcan and Hestia datasets are impressive and well-described. The types of emissions that are included and excluded from the Vulcan dataset are detailed: the Vulcan dataset focuses on energy-related fossil fuel emissions, thereby missing greenhouse gas emissions related to non-fossil fuel activities, such as fugitive/evaporative emissions and direct industrial process emissions from activities such as steel production (however, the text specifies that emissions from cement production are included). It may be worth mentioning at this point that $CH_4$ and $N_2O$ are not included (this is mentioned on p24, but could be mentioned earlier – for example, on p6 the inclusion of carbon monoxide is discussed, which could be a good place to put information about treatment of $CH_4$ and $N_2O$). Waste management is an important part of traditional city inventories, and the urban planning/policy-making crowd may want to know if emissions from waste decomposition/incineration are included/excluded from the dataset.

7. The analysis of spatial clustering and local emissions "hotspots" is very interesting and could potentially have direct policy/planning relevance. While not necessarily brand-new information (high traffic flow and congestion are likely well known), the addition of the emissions consequences could potentially open the door to new forms of carbon-related financing/policy mechanisms.

8. Something that stands out in the abstract is the 10% difference in the model results and the planning authority's GHG inventory. However, in the paragraphs on p24, a bit more information is needed about the SCAG model to understand the comparison to Hestia and its relevance. SCAG is described as "a regional greenhouse gas emissions inventory for a base year period of 1990-2009 with projections out to the year 2035." Are these both direct emissions inventories? Does SCAG use a traditional accounting approach similar to the GPC or an approach closer to Hestia's methodology? Is the purpose of comparison for validation of Hestia, or to demonstrate the drawbacks in the SCAG inventory? A couple more lines about the SCAG methodology and the purpose of the comparison will help clarify.

9. In my opinion, the most important paragraph for policy makers is p30, lines 700-713. Policy makers/planner may be tempted to think, "We do these GHG inventories already, why should we look at this tool?" That paragraph directly answers the question of why an urban planner/policy maker should care about this work. I highly recommend bringing elements of this paragraph to the introduction and/or abstract. Perhaps a few lines in the introduction that are specifically directed at city planners and policy makers. I completely agree with the line, "The most important attribute of the Hestia-LA approach, therefore, is the potential it offers for targeting urban CO2 reduction policy more efficiently." This hook deserves an earlier appearance in the paper.

Technical Corrections

- As mentioned above, the two citations on p2, line 64, do not appear in the reference list - Typo in y-axis label of Figure 10b.

---

## Author Comment (AC1) · 21 Apr 2019

Thank you for the helpful review comments. We agree that a broader sweep of work in other cities would provide more context and a better understanding of the topic for the reader. We will include text and citations to many of the cities mentioned in the review and possible a few others, nothing those that have integrated bottom-up/top-down and those that haven't. A good place for additional text would be page 3, line 107. We will insert a new paragraph prior to "The Hestia approach has been used...."

---

## Author Comment (AC2) · 22 Apr 2019

We thank reviewer 2 for helpful comments and suggestions. We will make all corrections as suggested by the reviewer particularly those that recommend bringing some of the policy-relevance of the results to locations earlier in the text (perhaps in summary form). We will ensure that jargon terms are defined and making it clear what elements are not included in this emissions data product (other GHGs, waste management, etc).

---

## Author Response (AR1)

**Reviewer 1**

This paper describes the development of a high space and time fossil fuel CO2 in- ventory for Los Angeles. It is an important contribution to the literature. The paper covers the description of the data set well. The one thing I would recommend is that the authors take a broader view of the community they are trying to influence with their ideas. The introduction is very self-referential and pays little to no attention to CO2 observing/modeling projects in Paris, Zurich, San Francisco, Boston and likely others. Helping the uninitiated reader to understand how this project relates to others should be a goal of the introduction and would make the paper more influential.

[[author]] greater context is achieved by adding text with references to a few of the cities other than those in which the author has been involved. These examples attempt to include those engaged in top-down/bottom-up convergent efforts. I could find no peer-reviewed information about $CO_2$ observing/modeling efforts in Zurich. The added text is:

"For example, ongoing efforts at integration of atmospheric measurements and bottom-up emissions information are taking place in Paris (Breon et al., 2015; Staufer et al., 2016), Boston (Sargent et al., 2018), Salt Lake City (Mitchell et al., 2018) and London (Font et al., 2015), to name a few.

**Reviewer 2**

General Comments

1. The Hestia data products provide bottom-up fossil fuel emissions at the urban scale with
building/street and hourly space-time resolution. The data feed into atmospheric CO2 inversion
studies and can help guide to climate change mitigation options, includ- ing disaggregation of
national goals/policies to the local level. This paper focuses on the Los Angeles Megacity, the
Combined Statistical Area that includes Los Angeles, Orange, Riverside, San Bernardino, and
Ventura counties, providing CO2 emissions results at a spatial resolution of 1km by 1km for the
years 2010-2015. The input data are from the Vulcan Project, adjusted where superior local and
downscaled data are available. It was found that the study area emitted 48.06 MtC/yr. Of note,
Hestia emissions were found to be 10.7% larger than the estimate by the local metropolitan
planning agency, the Southern California Association of Governments (SCAG).

2. The Hestia-LA data product, and the Vulcan dataset in general, are exciting advances in the
urban greenhouse gas emissions quantification field. In particular, the data product goes beyond
traditional carbon inventories used by city planning agencies to include a higher spatial and
temporal resolution. This makes the data product particularly exciting for the development of
highly impactful and targeted carbon reduction policies in cities.

Specific Comments

3. This reviewer has a keen interest in urban planning and policy-related efforts for greenhouse
gas emissions reduction in cities. As such, the comments are intended to help improve the
paper's accessibility to a planning/policy audience and help specify the policy-related
importance of the Hestia-LA work.

4. This work is of interest to a variety of readers, including policy makers and urban planning
professionals. With that in mind, I would recommend defining a few of the atmospheric-science-
specific words and phrases earlier on in the introduction, particularly "flux" and "inversion",
ideally in the context of other phrases that are more recognizable to the policy maker or
layperson (e.g., "emissions", "CO2 emissions"). For example, the term "flux" begins to appear
regularly on page 3 (e.g., "flux measurements", "flux estimation", "surface fluxes"), and it could
be confusing to a reader who isn't used to the terminology. This would make the content more
accessible to policy audiences.

[[author]] We have clarified the terms mentioned. On lines 86-87, we provide a definition of
"flux":

"….ground-based eddy flux (i.e. emissions of $CO_2$ into the atmosphere and/or $CO_2$ being removed from the
atmospheric by vegetation) measurements…"

On lines 132-134, we define the term "inversion":

"….atmospheric $CO_2$ inversion (i.e. an approach whereby $CO_2$ concentration measurements in the atmosphere are
combined with models of wind motions to infer what the emissions emanating from the surface must be)."

The importance of the policy application has been emphasized in the abstract by flipping the
order of the application to inversion and the application to policy.

5. The introduction does a good job in describing the body of work on greenhouse gas
accounting in the urban environment, including the gap in existing methods that the Hestia
Project aims to address. The discussion of policy-related issues on p2-3 are good: lines 50-55

(contributions from city-based policies to meeting national/global commitments), 64-68 (data
and aggregation difficulties), 95-99 (translation to urban mitigation efforts). I would have liked
to see a few more lines on the benefits and drawbacks of policy-related/traditional greenhouse
gas emissions inventories (mentioned on p2, line 64 - note that these two citations do not actually
appear in the reference list). It would be good to reference the Global Protocol for Community-
scale GHG Emissions (GPC) here, as this is the current standard for city-based GHG inventories
in the policy world, used by both the C40 and GCoM (two organizations that are currently
mentioned in the paper). See https://ghgprotocol.org/greenhouse-gas-protocol- accounting-
reporting-standard-cities. The paragraph on page 29, lines 672-683, does an excellent job of
describing the issues with traditional urban inventories and the benefit of the Hestia approach.
Could a few aspects of this description appear in the introduction?

[[author]] The missing references have been added to the reference section. The Fong et al.
reference is to the GPC. The introduction now has some of the conceptual material that the
reviewer notes, previously found later in the paper.

Text has been added on lines 82-87 that clarifies the potential benefit of the Hestia-style
approach:

"The need for greater granularity and specificity of emissions promises more efficient policy solutions. As all cities
reach beyond the existing "low hanging fruit" of emissions mitigation (i.e. those actions that are already planned for
other reasons, those that are simple and cost-plus), competition for limited resources and policy justification will
increase. Having information that can isolate the most efficient and effective emission reduction investments
(specific roadways/intersections, building subdivisions or commercial building clusters), will be at a premium."

6. The data collection and processing effort involved in creating the Vulcan and Hestia datasets
are impressive and well-described. The types of emissions that are included and excluded from
the Vulcan dataset are detailed: the Vulcan dataset focuses on energy-related fossil fuel
emissions, thereby missing greenhouse gas emissions related to non-fossil fuel activities, such as
fugitive/evaporative emissions and direct industrial process emissions from activities such as
steel production (however, the text specifies that emissions from cement production are
included). It may be worth mentioning at this point that CH4 and N2O are not included (this is
mentioned on p24, but could be mentioned earlier – for example, on p6 the inclusion of carbon
monoxide is discussed, which could be a good place to put information about treatment of CH4
and N2O). Waste management is an important part of traditional city inventories, and the urban
planning/policy-making crowd may want to know if emissions from waste
decomposition/incineration are included/excluded from the dataset.

[[author]]: lines 141-142 now contains the following clarification:

"Emissions considered here are carbon dioxide only; other important greenhouse gases such as methane ($CH_4$) and
nitrous oxide ($N_2O$) are not included."

Line 167 now contains the additional clarification:

"Similarly emissions associated with waste decay (organic or inorganic) are not included."

7. The analysis of spatial clustering and local emissions "hotspots" is very interesting and could
potentially have direct policy/planning relevance. While not necessarily brand-new information
(high traffic flow and congestion are likely well known), the addition of the emissions
consequences could potentially open the door to new forms of carbon-related financing/policy
mechanisms.

8. Something that stands out in the abstract is the 10% difference in the model results and the
planning authority's GHG inventory. However, in the paragraphs on p24, a bit more information
is needed about the SCAG model to understand the comparison to Hestia and its relevance.
SCAG is described as "a regional greenhouse gas emissions inventory for a base year period of
1990-2009 with projections out to the year 2035." Are these both direct emissions inventories?
Does SCAG use a traditional accounting approach similar to the GPC or an approach closer to
Hestia's methodology? Is the purpose of comparison for validation of Hestia, or to demonstrate
the drawbacks in the SCAG inventory? A couple more lines about the SCAG methodology and
the purpose of the comparison will help clarify.

[[author]]: agreed. On line 584-585, we redraft the sentence to read:

"There are very few estimates that can serve as an assessment of the accuracy of the Hestia FFCO$_2$ emissions as few
inventory efforts have been accomplished at the sub-state spatial scale in the United States."

On lines 595-597, we add the sentence:

"We use only the reported scope 1 emissions which were based on the approach adopted by CARB based on
guidelines from the Intergovernmental Panel on Climate Change (CARB, 2010)."

9. In my opinion, the most important paragraph for policy makers is p30, lines 700- 713. Policy
makers/planner may be tempted to think, "We do these GHG inventories already, why should we
look at this tool?" That paragraph directly answers the question of why an urban planner/policy
maker should care about this work. I highly recommend bringing elements of this paragraph to
the introduction and/or abstract. Perhaps a few lines in the introduction that are specifically
directed at city planners and policy makers. I completely agree with the line, "The most
important attribute of the Hestia- LA approach, therefore, is the potential it offers for targeting
urban CO2 reduction policy more efficiently." This hook deserves an earlier appearance in the
paper.

[[author]] some of the themes noted have been brought forward to lines 82-87. For example:

"The need for greater granularity and specificity of emissions promises more efficient policy solutions. As all cities
reach beyond the existing "low hanging fruit" of emissions mitigation (i.e. those actions that are already planned for
other reasons, those that are simple and cost-plus), competition for limited resources and policy justification will
increase. Having information that can isolate the most efficient and effective emission reduction investments
(specific roadways/intersections, building subdivisions or commercial building clusters), will be at a premium."

Technical Corrections

- As mentioned above, the two citations on p2, line 64, do not appear in the reference list - Typo
in y-axis label of Figure 10b.

[[author]]: Missing references have been added to the reference section. Figure 10b has been
corrected.

[revised manuscript text omitted]